# Continuous Language Model Interpolation yields Dynamic and Controllable Text Generation

**Sara Kangaslahti** *sarakangaslahti@g.harvard.edu*
*School of Engineering and Applied Sciences*
*Harvard University*

**David Alvarez-Melis** *dam@seas.harvard.edu*
*Kempner Institute*
*Harvard University*

**Reviewed on OpenReview:** *https://openreview.net/forum?id=xD9Nu2Wah4*

## Abstract

As large language models (LLMs) have gained popularity for a variety of use cases, making them adaptable and controllable has become increasingly important, especially for user-facing applications. In particular, linear interpolation between model parameters forms the backbone for many recent approaches to adapting models to user preferences. While the existing literature on LLM adaptation primarily focuses on finding methods that optimize for some set of performance criteria or user preferences, here we instead seek to better understand and characterize the behavior of dense, continuous interpolation between models. Specifically, we use low-rank updates to fine-tune a base model to various different domains, yielding a set of anchor models with distinct generation profiles. Then, we use the weight updates of these anchor models to parametrize the entire (infinite) class of models contained within their convex hull. We empirically show that varying the interpolation weights yields predictable and consistent change in the model outputs with respect to all of the controlled attributes simultaneously. We find that there is little entanglement between most attributes and identify and discuss the pairs of attributes for which this is not the case. Our results suggest that parameter merging facilitates flexible model adaptation due to its predictable behavior within the full interpolation region.[1]

## 1 Introduction

Large language models (LLMs) are used for a diverse set of applications due to their high performance across a wide spectrum of tasks (Bubeck et al., 2023). In many common LLM use cases (such as chatbots), different users often have distinct and continuously evolving preferences for the type of output they want. For example, a user might want a creative and verbose response for certain queries, but a concise and precise response for others. In practice, a user may try different variations of the same query successively until they elicit a desired generation. This trial-and-error process can be time-consuming and lacks guaranteed results, especially since minor word changes in a prompt can have disproportionate impact on the output. Additionally, expressing fine-grained continuous preferences (e.g., simplifying a response by 25%) is often difficult in —inherently discrete— natural language. These challenges are exacerbated when the user has complex, multi-faceted preferences (e.g., a specific combination of simplicity, formality, and verbosity) that they expect the generation to satisfy all at once. As a result, it is critical to understand how to adapt LLMs to user preferences and constraints in a fine-grained and predictable way.

Prior work in controllable text generation (CTG) has largely focused on optimizing for one set of control criteria through techniques such as instruction tuning (Zhou et al., 2023), modifying the output probability

---

[1]Code: https://github.com/skangasl/continuous-lm-interpolation

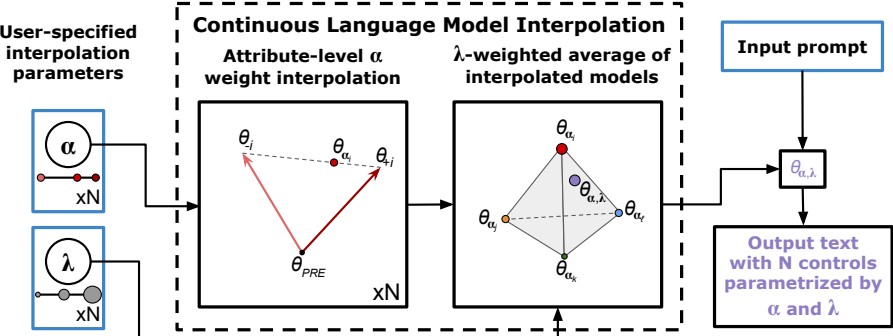

Figure 1: **Overview of our continuous model interpolation framework.** Given a collection of 'anchor' models fine-tuned on datasets at opposite ends of an attribute spectrum (e.g., $\theta_{+i}$: positive and $\theta_{-i}$: negative sentiment) for $N$ different attributes, the user selects interpolation parameters $\alpha$ (per-attribute spectrum modulation) and $\lambda$ (attribute mixture weights), which are used to generate a model with weights $\theta_{\alpha,\lambda}$ tailored to that specific parameter choice. We use this framework to analyze the behavior of interpolated models within the entire continuous region between the anchor models.

distributions (Pascual et al., 2021; Yang & Klein, 2021; Dekoninck et al., 2024), changing model activations at inference time (Li et al., 2023), learning modifications to the embeddings (Li & Liang, 2021; Han et al., 2023), or training (Keskar et al., 2019; Krause et al., 2021). These methods, however, are not parametrized continuously and instead require a fixed set of controls criteria. Thus, to achieve fine-grained control in the range between different attribute classes, they would have to be individually run for each specific set of intermediate attribute values, which is prohibitively expensive over a continuous range. Similarly, fine-tuning models with data that contains a proportionate amount of documents from each desired objective (ie 0.5 positive and 0.5 negative sentiment documents for a neutral model) would allow for the most precise optimization. However, this is computationally infeasible to do for each combination of control variables and strengths of control in the entire (infinite) set of possible combinations.

With these challenges in mind, linear weight interpolation has been proposed to adapt LLMs in a manner that takes advantage of the strengths of fine-tuning while making it computationally feasible to dynamically adapt the model for specific tasks or user preferences. Recent work has demonstrated that multiple pre-trained or fine-tuned models can be effectively composed through linear weight interpolation (Wortsman et al., 2022; Ilharco et al., 2023). This has also been shown to extend to models trained with parameter-efficient fine-tuning (PEFT) methods (Zhang et al., 2023; Huang et al., 2024) such as low-rank adaptation (Hu et al., 2021). As a result, linear interpolation has been used to adapt models through improving multitask performance (Matena & Raffel, 2021; Yadav et al., 2023; Ortiz-Jimenez et al., 2023) or aligning models to user preferences in the reinforcement learning setting (Ramé et al., 2023; Jang et al., 2023; Wang et al., 2024). However, these approaches largely center on optimizing the interpolation for task performance or user reward, so the behavior of fine-grained interpolation in the full continuous region between models remains poorly understood.

In this work, we show that linear weight interpolation effectively provides a continuous parametrization of the (infinite) 'convex hull' of a set of fine-tuned models. To do so, we fine-tune two endpoint anchor models for each control attribute, one at each extreme of attribute strength. We then interpolate along the vector between the weights of these two models for each attribute before computing a weighted average across all of the single-attribute interpolated models (Figure 1). Thus, varying the interpolation and averaging weights gives us *dense coverage* of the parameter space between endpoint models. We evaluate linear weight interpolation for multiple style attributes and demonstrate empirically that changes in the interpolation and averaging weights yield predictable and consistent responses in each control attribute in the generations.

A potential pitfall of linear interpolation is that, as seen in prior work in the vision domain (Ortiz-Jimenez et al., 2023), the weights for different single-attribute interpolated models may be entangled. This could lead to unexpected correlations between attributes in the averaged models. These correlations are detrimental to controllability, as changing the interpolation weights for one attribute could have an unexpected effect on

the correlated attributes in the output text. However, we find that there is surprisingly little entanglement between the vast majority of control attributes and analyze the pairs of controls where this is not the case.

In summary, our key contributions are: (1) we provide a framework for analyzing parameter-efficient adaptation in the continuous interpolation region between models fine-tuned with various distinct generation objectives; and (2) we demonstrate that changes in the interpolation yield smooth and predictable changes in the properties of the generated text across multiple sets of controls with limited entanglement.

## 2 Fine-tuning and Weight Interpolation

We evaluate the ability of weight interpolation to control the outputs of LLMs on five commonly used style attributes defined in prior style transfer literature (Jin et al., 2022): simplicity, formality, politeness, sentiment, and humor. For every style characteristic, we first fine-tune two endpoint 'anchor' models, each of which optimizes for one extreme of the style attribute. We then use these models as the basis of the interpolation scheme.

### 2.1 Datasets

For each style attribute, we fine-tune a separate anchor Llama2-7b model (Touvron et al., 2023) on two English datasets representing the extremes of the attribute level. For simplicity, we use the TinyStories dataset (Eldan & Li, 2023) to fine-tune a simple model and novel chapters from the BookSum dataset (Kryscinski et al., 2021) to fine-tune a complex model. We use the documents classified as formal and informal in Grammarly's Yahoo Answers Formality Corpus (GYAFC) dataset (Rao & Tetreault, 2018) to fine-tune formal and informal models. For the politeness attribute, we use the documents in the highest and lowest politeness class in the work by Madaan et al. (2020) for fine-tuning polite and impolite models, respectively. We fine-tune positive and negative sentiment models using the Stanford Sentiment Treebank (SST-2) dataset (Socher et al., 2013). For humor, we use the FlickrStyle dataset (Gan et al., 2017) to fine-tune humorous and non-humorous models.

### 2.2 Fine-tuning

We fine-tune our models in a parameter-efficient manner using Low-Rank Adaptation (LoRA, Hu et al., 2021), which keeps pretrained model weights frozen but learns an additive low-rank matrix update for each layer during fine-tuning. Denoting the pretrained language model weights as $\theta_{PRE} \in \mathbb{R}^{d_1 \times d_2}$, LoRA computes the updated weights as:

$$\theta = \theta_{PRE} + BA \tag{1}$$

Here, $A \in \mathbb{R}^{k \times d_2}$ and $B \in \mathbb{R}^{d_1 \times k}$ (with $k \ll d_1, d_2$) are trainable parameters learned during fine-tuning. We use LoRA as an adaptation method because it requires significantly fewer parameters than traditional fine-tuning while maintaining similar performance, so LoRA weights can be quickly modified and applied to large pretrained language models. We use the parameters in Appendix A.1 for fine-tuning the models and fine-tune two LoRA models per style characteristic, one on each of the extreme classes outlined in 2.1. We denote the two LoRA fine-tuned endpoint anchor models for attribute $i$ by $\theta_{+i} = \theta_{PRE} + B_{+i}A_{+i}$ and $\theta_{-i} = \theta_{PRE} + B_{-i}A_{-i}$.

### 2.3 Linear weight interpolation

Given a collection of fine-tuned model weights obtained by LoRA as described above, we generate interpolated models by linearly interpolating between their weights. We formulate linear weight interpolation between the LoRA fine-tuned models in terms of interpolation weights $\alpha_i$ and attribute mixing weights $\lambda_i$ as shown in Figure 1. For a single attribute, we interpolate along the vector between the two fine-tuned endpoint models by computing

$$\begin{aligned}\theta_{\alpha_i} &= \alpha_i \theta_{+i} + (1 - \alpha_i)\theta_{-i} \\ &= \theta_{PRE} + \alpha_i B_{+i}A_{+i} + (1 - \alpha_i)B_{-i}A_{-i}\end{aligned} \tag{2}$$

We call $\alpha_i$ the interpolation weight for the $i$th attribute dimension. We note that $\alpha_i = 0$ and $\alpha_i = 1$ correspond to letting the interpolated model equal the fine-tuned models $\theta_{\alpha_i} = \theta_{-i}$ and $\theta_{\alpha_i} = \theta_{+i}$, respectively. Using Equation 2, we then combine multiple interpolated models $\theta_{\alpha_i}$ by taking their weighted sum:

$$\theta_{\alpha,\lambda} = \sum_i \lambda_i \theta_{\alpha_i} \tag{3}$$

We denote $\lambda_i$ to be the mixing weight for the $i$th attribute and constrain $\sum_i \lambda_i = 1$. We note that the case with one attribute dimension corresponds to the sum having a single term with $\lambda_1 = 1$. With this formulation, we can construct any model in the convex hull of the fine-tuned models by choosing appropriate interpolation weights $\alpha$ and mixing weights $\lambda$. While the raw interpolation parameters do not have a clear meaning, we seek to show that a user can controllably increase or decrease the level of each attribute by changing $\alpha$ and $\lambda$.

### 2.4 Evaluation

To evaluate the interpolated models, we use a subset of 1k randomly sampled prompts from the WritingPrompts dataset (Fan et al., 2018) and generate 3 continuations for each prompt. We compute scores for each attribute to evaluate the level of the control criterion. Similarly to prior work on text style transfer (Xu et al., 2018), we fine-tune a RoBERTa (Liu et al., 2019) classification head on each attribute using a held out split of the datasets in 2.1 and compute a sigmoid over the output logits to obtain the probability of class 1, which we report as the attribute score. We label the documents such that an attribute score closer to 1 corresponds to a document that is more simple, formal, polite, positive in sentiment, or humorous. We also compute perplexity on the test split of the WikiText dataset (Merity et al., 2016) and n-gram diversity of the WritingPrompts generations to evaluate text quality.

**Baselines:** We provide a comparison of weight interpolation to two baselines: DExperts (Liu et al., 2021) and model arithmetic (Dekoninck et al., 2024), as these are the main prior approaches that do not use linear weight interpolation but allow for continuous values of attribute strength parameter. DExperts is formulated using a base model and two fine-tuned endpoint models. Then, given a prompt $x_{<t}$, if we denote the output logits of the base model at time $t$ as $z_t$ and the logits of the two endpoint models as $z_t^+$ and $z_t^-$, respectively, then the DExperts output probability distribution is defined by $P(X_t|x_{<t}) = \text{softmax}(z_t + \alpha(z_t^+ - z_t^-))$. The model arithmetic baseline uses the same formula for computing the output probability distribution, but instead of using two fine-tuned endpoint models, it uses two prompt-conditioned endpoint models to produce $z_t^+$ and $z_t^-$. As a result, both DExperts and weight interpolation require $2 *$ number dimensions fine-tuned endpoint models, while model arithmetic does not require any. However, weight interpolation uses a single inference pass, while both of these approaches require $2 *$ number dimensions $+ 1$ inference passes (Table 4).

We use the fine-tuned anchor models for DExperts and prompt-condition Llama-2-7b (see A.3 for details) for model arithmetic. For both comparisons, we compute the attribute scores for $\alpha \in [-2, 2]$. To provide a direct comparison to weight interpolation, we scale the $\alpha$ parameter such that the scaled $\alpha = 0$ and $\alpha = 1$ correspond to the models with attribute score equal to that of the fine-tuned endpoint models. We evaluate these interpolation methods by their proximity to the ground truth interpolated models, which are Llama-2-7b models fine-tuned with $\alpha$ fraction data from class 1 and $1 - \alpha$ fraction data from class 0.

## 3 Continuous Language Model Interpolation

We begin by investigating the linear interpolations between each pair of fine-tuned anchor models (3.1). We then extend this analysis to the convex hull of anchor models for multiple attributes (3.2).

### 3.1 Linear interpolation for a single attribute dimension

We first explore the effect of moving along the vector between a single pair of fine-tuned anchor models. We note that $\alpha = 0$ and $\alpha = 1$ correspond to the two fine-tuned anchor models, while $\alpha \in (0.0, 1.0)$ is an interpolation and $\alpha \in (-\infty, 0.0) \cup (1.0, \infty)$ is an extrapolation along the vector between the models.

**Linear interpolation:** Figure 2 shows the effect of $\alpha$ on attribute score when interpolating between each pair of fine-tuned anchor models. As $\alpha$ increases, there is a smooth and predictable increase in the attribute

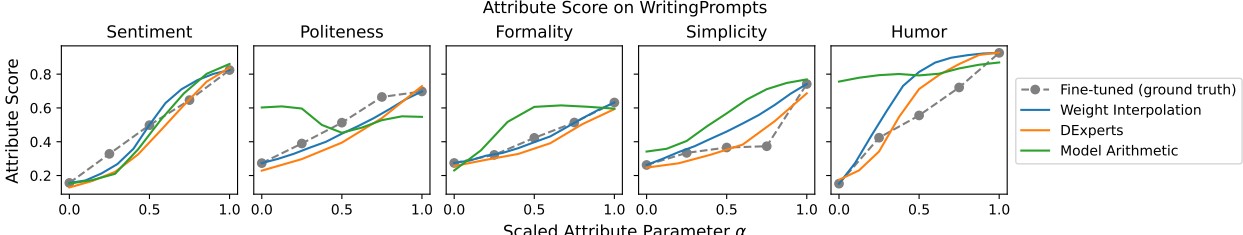

Figure 2: **Interpolated models recover custom fine-tuned models across the interpolation range**. We show the attribute scores for our interpolation framework with weight $\alpha$ compared to DExperts (Liu et al., 2021) and model arithmetic (Dekoninck et al., 2024) with $\alpha$ scaled such that the scaled $\alpha = 0$ and $\alpha = 1$ models have the same score as the fine-tuned endpoint models. Weight interpolation most closely follows the trend of the ground truth fine-tuned models. We find similar results for other language models in Appendix A.7.

Table 1: **Weight interpolation best approximates custom fine-tuned models while producing high-quality text**. For every attribute, we report the mean absolute error (MAE) between the attribute scores of the custom fine-tuned models and the models for each approach with corresponding $\alpha$ attribute parameters. We evaluate the text quality using WikiText perplexity and n-gram diversity scores. Weight interpolation produces text significantly closer in attribute score to the fine-tuned models with similar perplexity and better diversity than previous approaches. Appendix A.7 shows results for other language models and and we report additional diversity metrics in Table 6.

| | Attribute score mean absolute error (MAE) | | | | | | WikiText Perplexity | Diversity | | |
| --- | --- | --- | --- | --- | --- | --- | --- | --- | --- | --- |
| | Sentiment | Politeness | Formality | Simplicity | Humor | Average | | Dist-1 | Dist-2 | Dist-3 |
| Fine-tuned (ground truth) | - | - | - | - | - | - | 5.040 | 0.930 | 0.941 | 0.896 |
| DExperts (Liu et al., 2021) | 0.066 | 0.132 | 0.105 | 0.161 | **0.080** | 0.109 | 5.031 | 0.930 | 0.892 | 0.835 |
| Model arithmetic (Dekoninck et al., 2024) | 0.075 | 0.177 | 0.088 | 0.143 | 0.276 | 0.152 | **4.900** | 0.923 | 0.936 | 0.889 |
| Weight interpolation | **0.036** | **0.041** | **0.006** | **0.066** | 0.105 | **0.051** | 4.947 | **0.934** | **0.939** | **0.891** |

score for all of the control dimensions. Furthermore, linear interpolation follows the trend of the ground truth fine-tuned models more closely than either of the baselines while achieving similar perplexity and better diversity (Table 1). This demonstrates that interpolation can be used to approximate this continuous class of intermediate fine-tuned models by fine-tuning only 2 endpoint models. In contrast, for the model arithmetic baseline, the unpredictability of the attribute score in the politeness and humor dimensions suggests that prompt-conditioned models provide somewhat inconsistent control in the continuous interpolation setting in comparison to fine-tuned models. We find similar results for other language models (Appendix A.7) and diversity metrics (Table 6).

These results indicate that for one control attribute, weight interpolation between two endpoint models yields fine-grained control over the model outputs. Furthermore, the trend of increase with $\alpha$ appears linear in some cases. For the majority of the attribute dimensions (politeness, formality, and simplicity) we observe a linear increase in the score as $\alpha$ increases in the interpolation region. On the other hand, the other control dimensions (sentiment and humor) have a nonlinear increase in attribute score with $\alpha$ due to plateaus at the extremes.

**Linear extrapolation:** Figure 3 shows the attribute scores when extrapolating linearly beyond the two fine-tuned models along the vector between them. We find that even beyond the region of interpolation between the two fine-tuned models, there is a small stable extrapolation regime up to $\alpha$ values of around $-1$ and 2 (Figure 3). In this region, for many of the attributes, the attribute score continues to behave predictably as $\alpha$ is increased. However, beyond the stable extrapolation values, there is an unstable extrapolation regime where the attribute score changes unpredictably as $\alpha$ is varied. This is likely due to the model output quality degrading, since as shown in Figure 4 and Figure 11, the model perplexity increases sharply and the diversity decreases starting near the edges of the stable extrapolation regime. While prior work has shown that linear

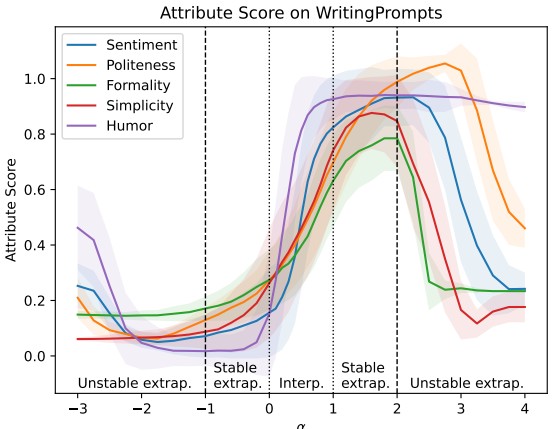

Figure 3: **Effect of linear weight extrapolation for a single attribute dimension.** For each style attribute, we report the attribute score when linearly extrapolating beyond the fine-tuned models ($\alpha < 0$ and $\alpha > 1$). There is a stable region where the score changes smoothly until a certain point (around $\alpha$ equal to $-1$ and 2), where performance degrades and the extrapolation is unstable.

Figure 4: **Wikitext perplexity of linearly interpolated and extrapolated models.** We report the average perplexity (lower is better) of each model from Figure 3 on the Wikitext test set. For the extrapolated models ($\alpha < 0.0$ and $\alpha > 1.0$), the perplexity increases rapidly beyond $\alpha$ values of around $-1$ and 2. We clip the y-axis at 7.0 for readability (full plot in Figure 10).

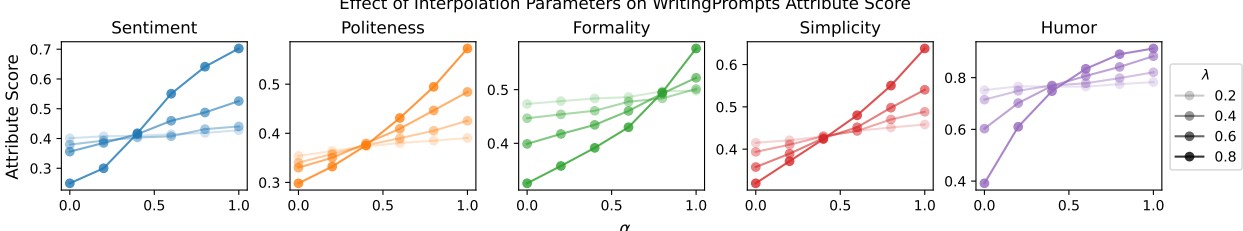

Figure 5: **Effect of $\alpha_i$ and $\lambda_i$ on 5-dimensional interpolation**. For each attribute, we show the attribute scores for models with the given $\alpha_i$ and $\lambda_i$ parameters, with all four other $\alpha_j = 0$ and $\lambda_j = (1 - \lambda_i)/4$. We find that increasing $\alpha_i$ consistently increases the attribute score and increasing $\lambda_i$ consistently increases the effect of $\alpha_i$.

weight extrapolation can be used for tasks such as model unlearning (Ilharco et al., 2023; Zhang et al., 2023), these results provide a cautionary tale against extrapolating too far, as they suggest that this ability only extends to a certain threshold before the attribute score and model outputs become unpredictable due to poor quality outputs. For the remainder of our experiments, we thus focus on the interpolation regime.

## 3.2 Multi-dimensional interpolation

In real-world LLM applications, users often have diverse output preferences across multiple control dimensions at once, and these preferences may change dynamically for different inputs to the LLM. In this section, we show that linear interpolation between fine-tuned parameter-efficient adapters can be used to parametrize a whole convex hull of models, which can be used to dynamically generate text with attribute levels specified on-the-fly.

### 3.2.1 Parametrization of the convex hull

**Analysis of interpolation parameter $\alpha$ and $\lambda$:** We find that when interpolating across up to five attribute dimensions, modifying the weight parameters $\lambda_i$ and $\alpha_i$ results in predictable, fine-grained control over

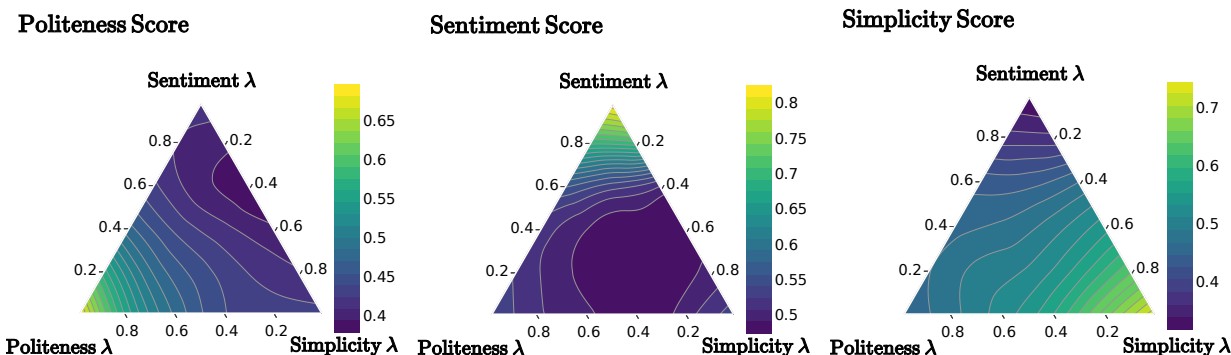

Figure 6: **Effect of $\lambda_i$ on interpolation between the sentiment, politeness, and simplicity dimensions for $\alpha_i = 1$.** The vertices of the triangle represent the models with $\alpha_i = 1$ for each of the three attribute dimensions. The scores in the simplex of $\lambda$ weights between the three control dimensions smoothly interpolate between the extreme models.

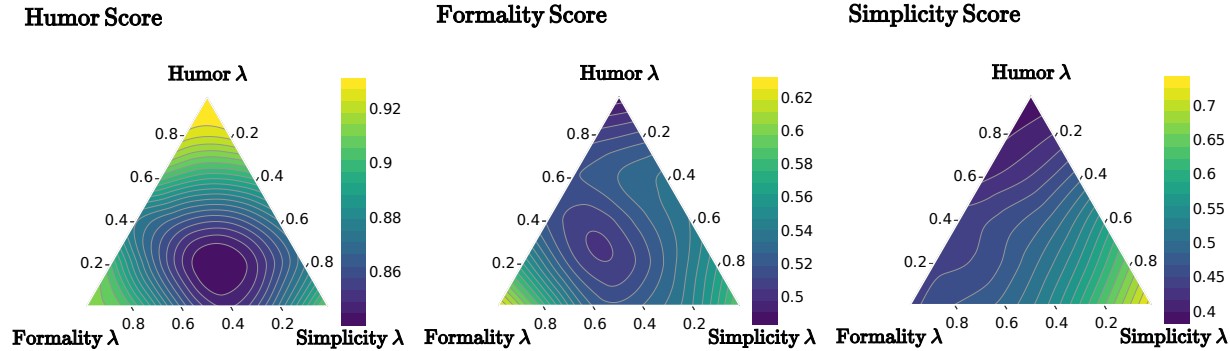

Figure 7: **Effect of $\lambda_i$ on interpolation between the humor, formality, and simplicity dimensions for $\alpha_i = 1$.** The vertices of the triangle represent the models with $\alpha_i = 1$ for each of the three attribute dimensions. The scores in the simplex of $\lambda$ weights smoothly interpolate between the three endpoints in the simplicity case, but the averaged models are the most neutral in the other cases due to correlations between the control dimensions.

the attribute scores for the desired attributes while having a comparatively small effect on the remaining attributes. Each attribute plot in Figure 5 shows that increasing the $\alpha_i$ parameter for interpolating between the fine-tuned models increases the attribute score for the $i$th attribute in a predictable manner. Similarly, as the model mixture parameter $\lambda_i$ increases, the effect on the attribute score of changing $\alpha_i$ increases.

**Changing mixing parameters $\lambda$ for multiple attributes at once:** We also analyze the relationship throughout the whole simplex of $\lambda$ weights for sets of three control dimensions in Figures 6 and 7 (as well as Appendix Figures 29-42). For each set of three attributes listed, these plots show the scores in the three dimensional simplex of mixing weights $\lambda$ for which $\sum_i \lambda_i = 1$. The value of the interpolation weight $\alpha_i$ for each of the attributes is equal to 1 in Figures 6 and 7, so increasing the $\lambda$ weight of each attribute should increase the attribute score. We find that surprisingly, there is very limited entanglement between the majority of the combinations of attributes (such as in Figure 6), likely because their LoRA weights are relatively orthogonal to each other (Appendix Figure 44). In these cases, we observe an approximately even increase in score as $\lambda_i$ for a given attribute dimension increases, regardless of the other $\lambda_j$ parameters.

However, in some cases, such as humor in the humor-formality-simplicity simplex and formality in the humor-formality-simplicity simplex with $\alpha_i = 1$ (Figure 7), we observe regions at the corners of the simplex that are close to the other fine-tuned models and have a high attribute score. This is because these other models are correlated with a positive attribute score, so the mixture of models is the most neutral model. Nevertheless, this still has a limited effect on the attribute score, since even in these cases with correlations,

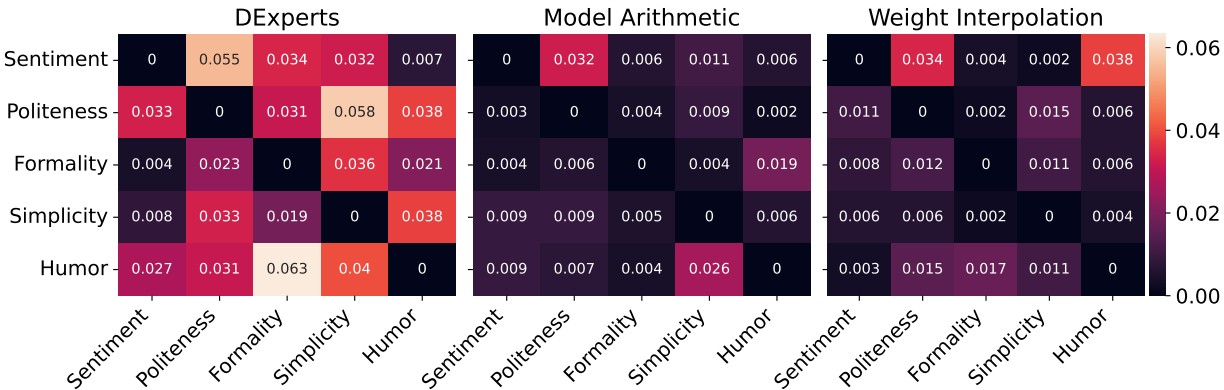

Figure 8: **Weight interpolation has entanglement lower than or comparable to prior approaches.** For each pair of dimensions, we fix (scaled) $\alpha_i = 1$ for the dimension in each row and vary (scaled) $\alpha_j$ between 0 and 1 for the dimension in each column. We set $\lambda_i = \lambda_j = 0.5$. Then, we report entanglement as the area under the curve of absolute value of change in attribute score as $\alpha_j$ increases. Weight interpolation is less entangled than DExperts (Liu et al., 2021) and has similar entanglement to model arithmetic (Dekoninck et al., 2024). These results hold across different language model sizes and families (Appendix A.9).

the score still has the expected behavior unless the mixing weight $\lambda_j$ is greater than around 0.4 to 0.6 for the correlated control dimensions. This indicates that in practice, the model has smoothly increasing attribute scores with $\lambda_i$ for all pairs of attributes when $\lambda_j$ for the other attribute dimensions remains sufficiently low.

These results demonstrate that as the parameters $\lambda_i$ and $\alpha_i$ are increased for the $i$th attribute, there is a significant effect on the attribute score for the $i$th control dimension and a limited effect on the scores for the remaining attributes. Therefore, $\lambda_i$ and $\alpha_i$ parametrize the convex hull of models between all of the attribute dimensions and yield fine-grained control over the model outputs with respect to all of the attributes being considered.

### 3.2.2 Entanglement analysis

Given the results from the simplex plots, we analyze the entanglement between each pair of dimensions to better understand the attribute score correlations. For every pair of dimensions $(i, j)$, if the two attributes are not entangled, it should be the case that if $\alpha_i$ is held constant, then changing $\alpha_j$ does not affect the attribute score. We can thus evaluate the amount of entanglement for a given $\alpha_j$ value by computing the absolute value of the difference between the attribute score for that $\alpha_j$ value and the attribute score when $\alpha_j = 0$. We do so for scaled $\alpha_i = 1$ (for $\alpha_i = 0$ see Appendix Figure 20 and for additional language models see Appendix A.9) and $\alpha_j \in \{0, 0.25, 0.5, 0.75, 1.0\}$ with $\lambda_i = \lambda_j = 0.5$. We then compute the area under the curve (AUC) of the entanglements for these $\alpha_j$ values in Figure 8. We find that the entanglement for weight interpolation is comparable to that of model arithmetic (Dekoninck et al., 2024), but less than that of DExperts (Liu et al., 2021).

## 4 Related Work

### 4.1 Controllable text generation (CTG)

As it is crucial to constrain generated text in many downstream applications, CTG has been a recent focus of NLP research. Methods such as CTRL (Keskar et al., 2019) and GeDI (Krause et al., 2021) pretrain language models on text prepended with control codes and generate text conditioned on the desired control. However, these methods require pretraining a new model if new controls are added, which is computationally expensive. To mitigate these issues, a variety of methods have been proposed to perform CTG without additional language model training. For example, Liu et al. (2021); Khalifa et al. (2021); Pascual et al.

(2021); Yang & Klein (2021); Dekoninck et al. (2024) constrain language model outputs by modifying their output probability distributions. Li & Liang (2021); Qian et al. (2022) learn prefixes and Dathathri et al. (2019); Han et al. (2023) train additional classifiers to guide generation. Subramani et al. (2022); Hernandez et al. (2023); Li et al. (2023); Turner et al. (2023) control model outputs by changing activations at inference time. Kumar et al. (2021) optimize the inference decoding. Mireshghallah et al. (2022); Qin et al. (2022) use energy-based constrained generation and Zhou et al. (2023) use instruction tuning for CTG.

However, we emphasize that the goal of our work is not to output text with the greatest or least possible amount of a given style attribute, but instead to show that interpolation can be used to dynamically and controllably increase or decrease the amount of a given style attribute. While most of these existing methods can be used to control text for multiple style attributes, the majority of them focus on binary or multi-class control (for example controlling text so that it has positive or negative sentiment) and would need to be re-run for each combination of attribute levels, which is not feasible for "continuous" adaptive control.

As a result, among existing methods only DExperts (Liu et al., 2021) and model arithmetic (Dekoninck et al., 2024) are composable and achieve fine-grained control over multiple attributes at once. Both weight interpolation and DExperts require fine-tuning two endpoint models for each attribute, while model arithmetic uses prompt-conditioned models. However, both DExperts and model arithmetic compose multiple models at inference time, so the inference cost is significantly higher than weight interpolation, especially as the model size and number of controlled attributes increases. In addition, our experiments show that weight interpolation provides more precise fine-grained control and less entanglement than these approaches over the range of intermediate models.

### 4.2 Weight interpolation

Our work builds on prior work on linear weight interpolation, such as task vectors (Ilharco et al., 2023), parameter-efficient task vectors (Zhang et al., 2023), and model souping (Wortsman et al., 2022), as we use linear interpolation and weighted model averaging as the basis for our analysis. Prior work in this domain has focused mainly on improving multitask performance when composing fully fine-tuned models (Matena & Raffel, 2021; Yadav et al., 2023; Ortiz-Jimenez et al., 2023) or parameter-efficient fine-tuned models (Huang et al., 2024; Jiang et al., 2024). However, these methods all differ from our work, since they focus on combining model weights to improve a single multitask objective rather than analyzing performance across a wide range of flexible, diverse objectives. These approaches are orthogonal to our work and could be used in conjunction with it to better combine the $\alpha$-interpolated models.

Beyond multitask performance, a variety of reinforcement learning approaches have been proposed for language model alignment to user preferences. Specifically, rewarded soups first fine-tunes a model for each preference and computes the combination of interpolation parameters that maximizes the user's reward on a validation set (Ramé et al., 2023), personalized soups uses a multi-objective reinforcement learning objective combined with parameter merging (Jang et al., 2023), and conditional language policy performs multi-objective fine-tuning to learn a set of parameters that is steerable by conditioning on the user reward at inference time (Wang et al., 2024). These methods show that reinforcement learning objectives can be combined with weight interpolation in order to optimize over a wide range of user preferences. However, they do not analyze the behavior across the full interpolation region and instead center around developing methods that obtain close to pareto-optimal reward or are more preferential to users with specified reward functions.

Perhaps most similar to our work are methods that analyze the interpolation regime between the weights of fine-tuned models over a range of outputs (Gandikota et al., 2023; Nylund et al., 2023). However, Gandikota et al. (2023) focus on the vision domain and use a fine-tuning objective specific to diffusion models, and Nylund et al. (2023) only analyze control over the time dimension.

## 5 Conclusion

In this work, we show that continuous linear interpolation between low-rank fine-tuned models can be used to parametrize the models in their convex hull. We achieve fine-grained, predictable control over multiple attributes of style at once by changing the interpolation weights between two anchor fine-tuned models and

the mixing weights between different interpolated attribute models. We find that the interpolation profiles between models are smooth and there is surprisingly little entanglement between the models for different control dimensions. In other words, changing the weight for one attribute has a very small effect on the scores for other attributes, especially for sufficiently small mixing weights. As a result, linear weight interpolation produces predictable behavior in the entire continuous space of models between the fine-tuned anchors.

## Future work

The main limitation of our work is that some pairs of attributes are correlated, so when a correlated model has a large mixing weight, it can unpredictably affect other control attributes. While there is lower or comparable correlation in weight interpolation as compared to other approaches, in the future it would be valuable to investigate whether this correlation is inherent to the pair of tasks or if it can be eliminated. For example, text that is more polite might always be more formal. However, it may be the case that some correlations can be reduced by regularizing the LoRA updates to be more orthogonal to each other or by merging the $\alpha$-interpolated using more sophisticated methods that have recently shown improvement over naive weight averaging in the multitask setting (Matena & Raffel, 2021; Yadav et al., 2023; Ortiz-Jimenez et al., 2023). Similarly to prior work on task vectors (Ilharco et al., 2023), correlations could also potentially be used to combine existing control attributes to make new ones (such as by combining irony and humor to control for sarcasm).

Another limitation is that the average generation attribute scores are limited to the range between the attribute scores of the fine-tuned anchor models. The single attribute extrapolation results could be expanded upon to better understand when extrapolation can be used to extend the range of the control attribute style. We also only consider controlling for text style in our experiments, but analyzing linear interpolation when controlling for other attributes such as topics is a possible direction for future work.

### Acknowledgments

SK is supported by the National Science Foundation Graduate Research Fellowship under Grant No. DGE 2140743. DAM and SK acknowledge support from the Kempner Institute, the Aramont Fellowship Fund, and the FAS Dean's Competitive Fund for Promising Scholarship.

## Broader Impacts

Continuous weight interpolation may output text that contains existing biases from the pre-trained models and fine-tuning datasets. It could also be used to control the level of undesirable attributes such as toxicity. However, we believe that this work is still beneficial overall, since we can better understand the behavior of linearly interpolation for adapting LLMs, and these issues are faced by all pre-trained and fine-tuned language models.

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

# A  Appendix

## A.1  Hyperparameters for fine-tuning

Table 2: **Parameters for LoRA fine-tuning.** We use 20 epochs for fine-tuning the sentiment attribute models and 1 epoch for the remaining fine-tuned models. All experiments were run on single NVIDIA A100 80GB SXM GPU nodes.

| LoRA hyperparameter | Value |
|---|---|
| Batch size | 64 |
| Learning rate | 5e-5 |
| LoRA $r$ | 32 |
| LoRA $\alpha$ | 16 |
| LoRA dropout | 0.1 |
| Max sequence length | 128 |
| Quantization | 4 bit |

Table 3: **Fine-tuning splits.** We report the number of examples from each attribute dataset used to fine-tune Llama2-7b generation and RoBERTa attribute scoring models. Each split is sampled from the combined train, test, and validation set.

| Domain | Llama2 split size | | RoBERTa split size |
|---|---|---|---|
| | Class 0 | Class 1 | |
| Sentiment Socher et al. (2013) | 25k | 30k | 10k |
| Politeness Madaan et al. (2020) | 78k | 100k | 20k |
| Formality Rao & Tetreault (2018) | 104k | 104k | 10k |
| Simplicity (Kryscinski et al., 2021; Eldan & Li, 2023) | 9k | 100k | 10k |
| Humor Gan et al. (2017) | 100k | 100k | 20k |

## A.2  Complexity comparisons to previous approaches

Table 4: **Comparing weight interpolation to previous CTG approaches.** N is the number of controlled attributes. Similarly to DExperts (Liu et al., 2021), weight interpolation requires fine-tuned anchor models. However, in contrast to prior work where the number of inference passes scales with the number of attributes, weight interpolation uses only a single inference pass.

| Approach | Anchor models | Inference passes |
|---|---|---|
| DExperts (Liu et al., 2021) | Fine-tuned | 2N + 1 |
| Model arithmetic (Dekoninck et al., 2024) | Prompt-conditioned | 2N + 1 |
| Weight interpolation | Fine-tuned | 1 |

## A.3  Model Arithmetic Formulation

For the model arithmetic (Dekoninck et al., 2024) comparison, we use the following formula, inspired by DExperts (Liu et al., 2021):

$$M + \alpha(M_{\text{pos}} - M_{\text{neg}})$$

Here, $M$ is the base model, $M_{\text{pos}}$ is the model conditioned for class 1, and $M_{\text{neg}}$ is the model conditioned for class 0. The system prompts used for conditioning are listed in Table 5.

Table 5: **System prompts used for conditioning model arithmetic.**

| Conditioned model name | Llama2-7b System Prompt |
| --- | --- |
| sentiment__pos | "The following is a positive story, with a very positive sentiment and a very positive tone." |
| sentiment__neg | "The following is a negative story, with very negative sentiment and a very negative tone." |
| formality__pos | "The following is a formal story, with very formal language and a very formal tone." |
| formality__neg | "The following is an informal story, with very informal language and a very informal tone." |
| simplicity__pos | "The following is a simple story, with very simple language." |
| simplicity__neg | "The following is a complex story, with very complex language." |
| humor__pos | "The following is a humorous story, with very humorous language and a very humorous tone." |
| humor__neg | "The following is a nonhumorous story, with factual language and a very serious tone." |
| politeness__pos | "The following is a polite story, with very polite language and a very polite tone." |
| politeness__neg | "The following is an impolite story, with very impolite language and a very impolite tone." |

## A.4  Attribute comparison to prompting

In this section (Figure 9), we provide comparisons for each attribute between prompting Llama2-13b-chat instruction-tuned models and the attribute score of models fine-tuned with $\alpha$ fraction of data from class 1 and $(1 - \alpha)$ fraction of data from class 0 for each attribute. We use the following prompting set-up inspired by Han et al. (2023):

- "Complete this story so that it embodies a sentiment score of 0.5, where 0 is negative and 1 is positive: "

- For each style attribute, we replace the words "sentiment", "negative", and "positive" with the corresponding attribute and class names, and 0.5 with the corresponding $\alpha$ score.

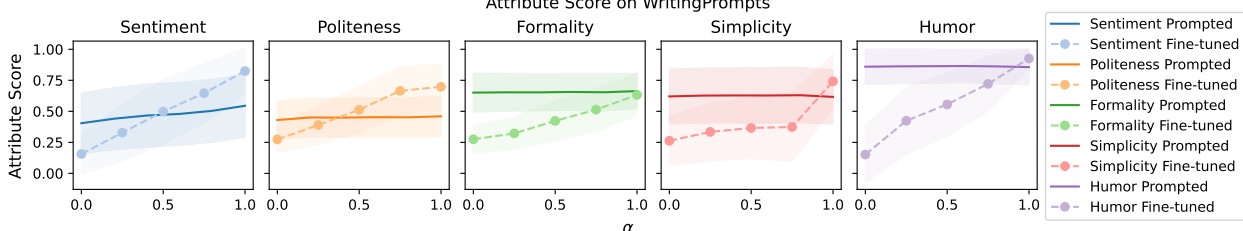

Figure 9: **Attribute scores for prompting Llama2-13b-chat versus fine-tuned models.** We report the attribute score when prompting Llama2-13b-chat models to produce output with score $\alpha$ as compared to the attribute score for models trained with $\alpha$ fraction class 1 and $1 - \alpha$ fraction class 0 data. The attribute scores of the prompted model only slightly increase with $\alpha$ and do not closely follow the trend of the fine-tuned models.

## A.5 Extrapolation text quality analysis

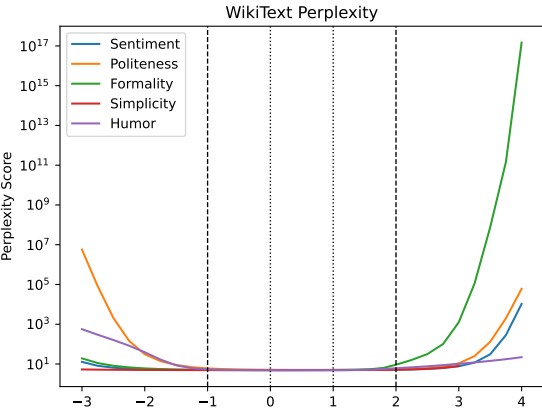

Figure 10: **Wikitext perplexity of linearly interpolated and extrapolated models.** We report the average perplexity of each model from Figure 3 on the Wikitext test set. For the extrapolated models not shown in Figure 4, the perplexity increases rapidly.

## A.6 Diversity analysis

Table 6: **Diversity comparison.** For each model, we report the mean compression ratio, BERT and ROUGE-L homogenization scores, and self-repetition score (lower is better for all scores) for the single-attribute linear interpolation from Table 1. Weight interpolation produces text with similar or better diversity than previous approaches for all of the models tested.

| Model | Method | Compression ratio | Homogenization score (BERT) | Homogenization score (ROUGE-L) | Self-repetition score |
|---|---|---|---|---|---|
| Llama-2-7b | Fine-tuned (ground truth) | 2.174 | 0.692 | 0.077 | 0.063 |
| | DExperts (Liu et al., 2021) | **1.459** | 0.699 | 0.086 | 0.207 |
| | Model Arithmetic (Dekoninck et al., 2024) | 2.542 | 0.692 | 0.100 | 0.627 |
| | Weight Interpolation | 2.140 | **0.690** | **0.073** | **0.036** |
| Llama-2-13b | Fine-tuned (ground truth) | 2.261 | 0.702 | 0.080 | 0.091 |
| | DExperts (Liu et al., 2021) | **2.209** | **0.684** | **0.070** | **0.051** |
| | Model Arithmetic (Dekoninck et al., 2024) | 2.598 | 0.694 | 0.101 | 0.728 |
| | Weight Interpolation | 2.238 | 0.698 | 0.080 | 0.065 |
| Llama-3.1-8B | Fine-tuned (ground truth) | 2.225 | 0.697 | 0.082 | 0.106 |
| | DExperts (Liu et al., 2021) | 2.402 | 0.689 | 0.097 | 0.228 |
| | Model Arithmetic (Dekoninck et al., 2024) | **2.170** | **0.689** | **0.081** | **0.048** |
| | Weight Interpolation | 2.208 | 0.695 | 0.081 | 0.080 |
| Qwen3-8B-Base | Fine-tuned (ground truth) | 2.269 | 0.706 | 0.083 | 0.090 |
| | DExperts (Liu et al., 2021) | 2.301 | **0.684** | **0.078** | 0.347 |
| | Model Arithmetic (Dekoninck et al., 2024) | 2.405 | 0.687 | 0.109 | 0.667 |
| | Weight Interpolation | **2.242** | 0.702 | 0.083 | **0.059** |
| Qwen3-14B-Base | Fine-tuned (ground truth) | 2.265 | 0.705 | 0.084 | 0.131 |
| | DExperts (Liu et al., 2021) | 2.317 | **0.683** | 0.089 | 0.556 |
| | Model Arithmetic (Dekoninck et al., 2024) | 2.417 | 0.688 | 0.103 | 0.944 |
| | Weight Interpolation | **2.233** | 0.702 | **0.084** | **0.072** |

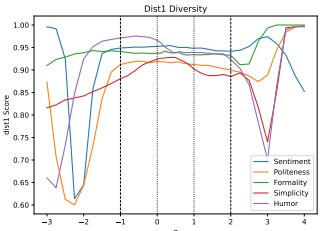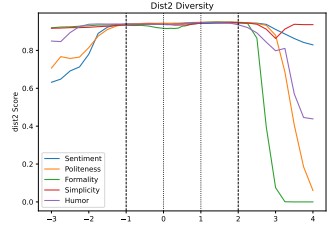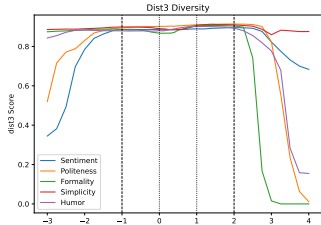

Figure 11: **N-gram diversity scores for the extrapolated models.** We report the 1-, 2-, and 3-gram diversity scores for the single-attribute interpolated and extrapolated models. The diversity scores remain similar to or between those of the endpoint fine-tuned models within the interpolation region ($\alpha \in [0,1]$) and in the stable extrapolation region ($\alpha \in [-1,0) \cup (1,2]$), but become unstable beyond the stable extrapolation region.

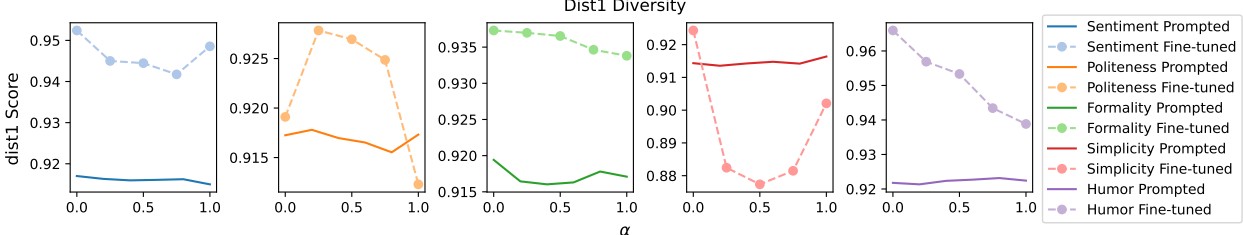

Figure 12: **1-gram diversity comparison of prompted versus fine-tuned models.** We report the dist1 diversity scores for the Llama2-13b-chat prompted models with weight $\alpha$ as compared to the perplexity for models trained with $\alpha$ fraction class 1 and $1 - \alpha$ fraction class 0 data. The diversity scores remain similar to the endpoint fine-tuned models within the interpolation region.

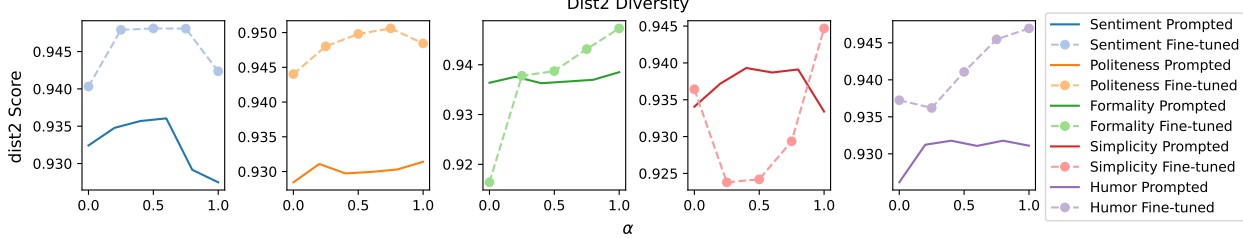

Figure 13: **2-gram diversity comparison of prompted versus fine-tuned models.** We report the dist2 diversity scores for the Llama2-13b-chat prompted models with weight $\alpha$ as compared to the perplexity for models trained with $\alpha$ fraction class 1 and $1 - \alpha$ fraction class 0 data. The diversity scores remain similar to the endpoint fine-tuned models within the interpolation region.

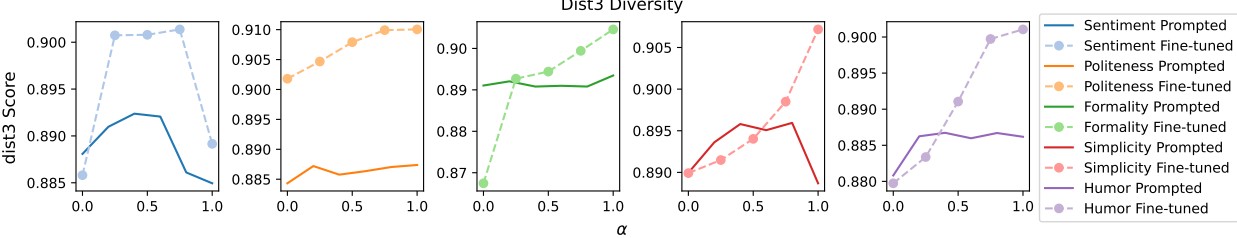

Figure 14: **3-gram diversity comparison of prompted versus fine-tuned models.** We report the dist3 diversity scores for the Llama2-13b-chat prompted models with weight $\alpha$ as compared to the perplexity for models trained with $\alpha$ fraction class 1 and $1 - \alpha$ fraction class 0 data. The diversity scores remain similar to the endpoint fine-tuned models within the interpolation region.

### A.7 Single attribute dimension interpolation results for additional models

We show that our single attribute dimension results generalize across language model sizes and families by reporting the results from Figure 2 and Table 1 for Llama-2-13b (Touvron et al., 2023), Llama-3.1-8B (Grattafiori et al., 2024), Qwen3-8B-Base (Yang et al., 2025), and Qwen3-14B-Base (Yang et al., 2025).

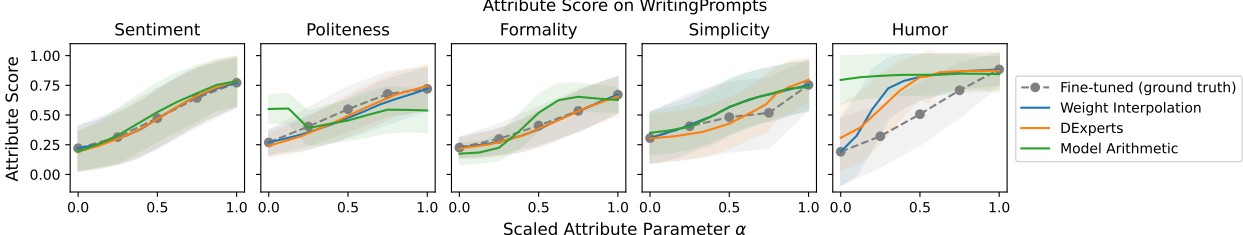

Figure 15: **Llama-2-13b interpolated models recover custom fine-tuned models across the interpolation range**. We show the attribute scores for our interpolation framework with weight $\alpha$ compared to DExperts (Liu et al., 2021) and model arithmetic (Dekoninck et al., 2024) with $\alpha$ scaled such that the scaled $\alpha = 0$ and $\alpha = 1$ models have the same score as the fine-tuned endpoint models. Weight interpolation most closely follows the trend of the ground truth fine-tuned models.

Table 7: **Llama-2-13b weight interpolation best approximates custom fine-tuned models while producing high-quality text**. For every attribute, we report the mean absolute error (MAE) between the attribute scores of the custom fine-tuned models and the models for each approach with corresponding $\alpha$ attribute parameters. We evaluate the text quality using WikiText perplexity and n-gram diversity scores. Weight interpolation produces text significantly closer in attribute score to the fine-tuned models with similar perplexity and better diversity than previous approaches. We report additional diversity metrics in Table 6.

| | Attribute score mean absolute error (MAE) | | | | | | WikiText Perplexity | Diversity | | |
|---|---|---|---|---|---|---|---|---|---|---|
| | Sentiment | Politeness | Formality | Simplicity | Humor | Average | | Dist-1 | Dist-2 | Dist-3 |
| Fine-tuned (ground truth) | - | - | - | - | - | - | 4.495 | 0.925 | 0.945 | 0.904 |
| DExperts (Liu et al., 2021) | 0.017 | 0.040 | 0.015 | 0.053 | 0.166 | 0.058 | 4.577 | 0.910 | 0.916 | 0.870 |
| Model arithmetic (Dekoninck et al., 2024) | 0.029 | 0.141 | 0.079 | 0.061 | 0.325 | 0.127 | **4.364** | 0.919 | 0.931 | 0.887 |
| Weight interpolation | **0.006** | **0.038** | **0.012** | **0.050** | **0.157** | **0.053** | 4.436 | **0.919** | **0.944** | **0.904** |

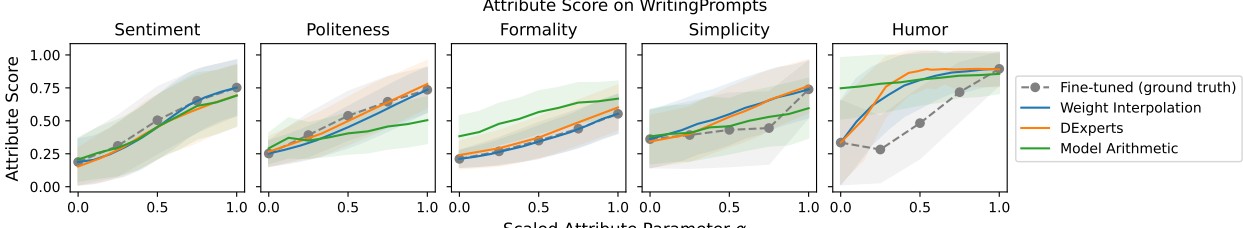

Figure 16: **Llama-3.1-8B interpolated models recover custom fine-tuned models across the interpolation range**. We show the attribute scores for our interpolation framework with weight $\alpha$ compared to DExperts (Liu et al., 2021) and model arithmetic (Dekoninck et al., 2024) with $\alpha$ scaled such that the scaled $\alpha = 0$ and $\alpha = 1$ models have the same score as the fine-tuned endpoint models. Weight interpolation most closely follows the trend of the ground truth fine-tuned models.

Table 8: **Llama-3.1-8B weight interpolation best approximates custom fine-tuned models while producing high-quality text**. For every attribute, we report the mean absolute error (MAE) between the attribute scores of the custom fine-tuned models and the models for each approach with corresponding $\alpha$ attribute parameters. We evaluate the text quality using WikiText perplexity and n-gram diversity scores. Weight interpolation produces text significantly closer in attribute score to the fine-tuned models with similar perplexity and diversity to previous approaches. We report additional diversity metrics in Table 6.

| | Attribute score mean absolute error (MAE) | | | | | | WikiText Perplexity | Diversity | | |
|---|---|---|---|---|---|---|---|---|---|---|
| | Sentiment | Politeness | Formality | Simplicity | Humor | Average | | Dist-1 | Dist-2 | Dist-3 |
| Fine-tuned (ground truth) | - | - | - | - | - | - | 4.495 | 0.917 | 0.946 | 0.917 |
| DExperts (Liu et al., 2021) | 0.048 | **0.025** | 0.034 | 0.072 | 0.198 | 0.076 | 4.577 | 0.876 | 0.923 | 0.876 |
| Model arithmetic (Dekoninck et al., 2024) | 0.037 | 0.127 | 0.182 | **0.058** | 0.281 | 0.137 | **4.364** | **0.917** | **0.952** | **0.917** |
| Weight interpolation | **0.018** | 0.040 | **0.003** | 0.078 | **0.171** | **0.062** | 4.436 | 0.910 | 0.942 | 0.910 |

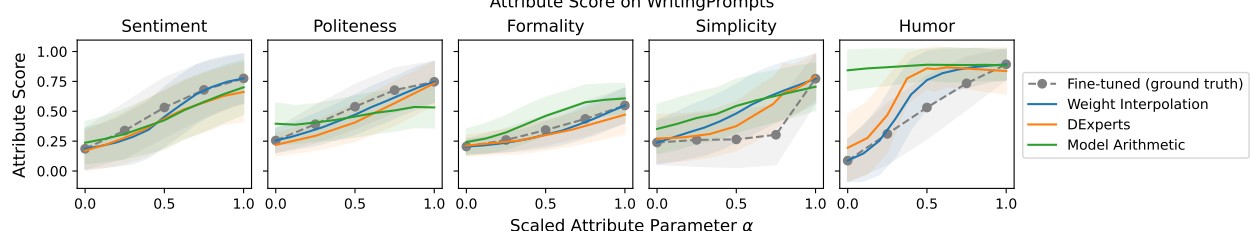

Figure 17: **Qwen3-8B-Base interpolated models recover custom fine-tuned models across the interpolation range**. We show the attribute scores for our interpolation framework with weight $\alpha$ compared to DExperts (Liu et al., 2021) and model arithmetic (Dekoninck et al., 2024) with $\alpha$ scaled such that the scaled $\alpha = 0$ and $\alpha = 1$ models have the same score as the fine-tuned endpoint models. Weight interpolation most closely follows the trend of the ground truth fine-tuned models.

Table 9: **Qwen3-8B-Base weight interpolation best approximates custom fine-tuned models while producing high-quality text**. For every attribute, we report the mean absolute error (MAE) between the attribute scores of the custom fine-tuned models and the models for each approach with corresponding $\alpha$ attribute parameters. We evaluate the text quality using WikiText perplexity and n-gram diversity scores. Weight interpolation produces text significantly closer in attribute score to the fine-tuned models with similar perplexity and better diversity than previous approaches. We report additional diversity metrics in Table 6.

| | Attribute score mean absolute error (MAE) | | | | | | WikiText Perplexity | Diversity | | |
|---|---|---|---|---|---|---|---|---|---|---|
| | Sentiment | Politeness | Formality | Simplicity | Humor | Average | | Dist-1 | Dist-2 | Dist-3 |
| Fine-tuned (ground truth) | - | - | - | - | - | - | 6.857 | 0.916 | 0.949 | 0.914 |
| DExperts (Liu et al., 2021) | 0.078 | **0.078** | 0.040 | **0.096** | 0.155 | 0.089 | 6.711 | 0.886 | 0.931 | 0.899 |
| Model arithmetic (Dekoninck et al., 2024) | 0.072 | 0.124 | 0.083 | **0.194** | 0.367 | 0.168 | **6.340** | 0.866 | 0.936 | 0.908 |
| Weight interpolation | **0.032** | 0.035 | **0.017** | 0.129 | **0.078** | **0.058** | 6.464 | **0.911** | **0.948** | **0.914** |

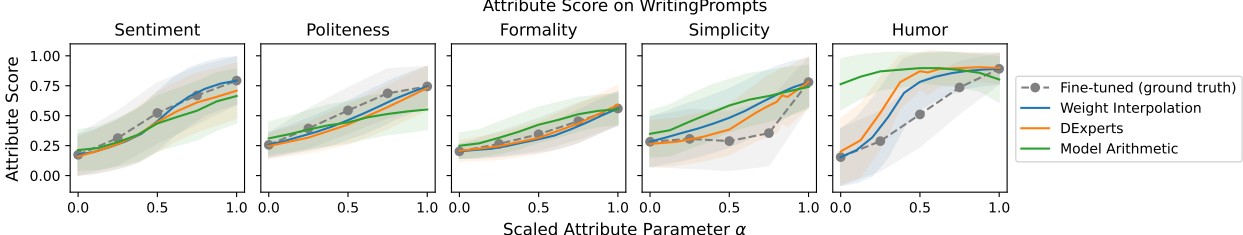

Figure 18: **Qwen3-14B-Base interpolated models recover custom fine-tuned models across the interpolation range**. We show the attribute scores for our interpolation framework with weight $\alpha$ compared to DExperts (Liu et al., 2021) and model arithmetic (Dekoninck et al., 2024) with $\alpha$ scaled such that the scaled $\alpha = 0$ and $\alpha = 1$ models have the same score as the fine-tuned endpoint models. Weight interpolation most closely follows the trend of the ground truth fine-tuned models.

Table 10: **Qwen3-14B-Base weight interpolation best approximates custom fine-tuned models while producing high-quality text**. For every attribute, we report the mean absolute error (MAE) between the attribute scores of the custom fine-tuned models and the models for each approach with corresponding $\alpha$ attribute parameters. We evaluate the text quality using WikiText perplexity and n-gram diversity scores. Weight interpolation produces text significantly closer in attribute score to the fine-tuned models with similar perplexity and better diversity than previous approaches. We report additional diversity metrics in Table 6.

| | Attribute score mean absolute error (MAE) | | | | | | WikiText Perplexity | Diversity | | |
| | Sentiment | Politeness | Formality | Simplicity | Humor | Average | | Dist-1 | Dist-2 | Dist-3 |
|---|---|---|---|---|---|---|---|---|---|---|
| Fine-tuned (ground truth) | - | - | - | - | - | - | 6.282 | 0.916 | 0.950 | 0.914 |
| DExperts (Liu et al., 2021) | 0.057 | 0.067 | **0.021** | **0.070** | 0.162 | 0.075 | 6.062 | 0.879 | 0.931 | 0.899 |
| Model arithmetic (Dekoninck et al., 2024) | 0.083 | 0.106 | 0.049 | 0.171 | 0.363 | 0.154 | **5.782** | 0.871 | 0.934 | 0.905 |
| Weight interpolation | **0.028** | **0.041** | 0.022 | 0.110 | **0.104** | **0.061** | 5.906 | **0.911** | **0.949** | **0.915** |

## A.8 Generation example

We provide an example generation to compare between weight interpolated models for a single attribute and prompting an instruction-tuned model (Llama2-13b-chat). We provide the model generations for the following prompt set-up inspired by Han et al. (2023):

- "Complete this story so that it embodies a sentiment score of 0.5, where 0 is negative and 1 is positive: You find a rip in time walking through the alleys . You enter it to find yourself "

- For each style attribute, we replace the words "sentiment", "negative", and "positive" with the corresponding attribute and class names, and 0.5 with the corresponding $\alpha$ score.

- We report the output until the first occurrence of a newline character or the amount of output that fits in 2-3 lines of the table.

In general, it is challenging to achieve fine-grained control over the output attributes with prompting as compared to interpolation. The prompted model often does not properly account for the $\alpha$ value and produces outputs at one attribute extreme or the other regardless of $\alpha$. Furthermore, for dimensions that are less commonly used in CTG (ie formality), the prompted model often produces very similar outputs for each value of $\alpha$, as reflected by many of the scores in Figure 9.

Table 11: **Generation comparison:** we present a comparison of generations for single attribute interpolation versus prompted Llama2-13b-chat for various $\alpha$ values with the prompt "You find a rip in time walking through the alleys . You enter it to find yourself "

| Dimension | $\alpha$ | Single attribute interpolation | Prompted Llama2-13b-chat |
|---|---|---|---|
| Sentiment | 0.0 | 40 minutes later still wondering what the h*ll you did wrong. [...] | months in the future. Everything has changed but... [...] |
| | 0.3 | 12 hours earlier with your hopes and sanity battered only to discover the time rip still [...] | months in the future. Everything has changed but it seems the world has gotten better. [...] |
| | 0.7 | 40 years earlier, passing through an archway into a deeply familiar but different world. [...] | 10 minutes in the past, before you were born. You decide to go back in time and give your younger self [...] |
| | 1.0 | 10 years older and wondering how it happened. [...] | 20 years earlier, in a world before the wars, global warming and the division of society. [...] |
| Politeness | 0.0 | 100 miles away from the starting line, but who told you to quit. [...] | 30 years earlier. You ask the current you what to do next to maximize [...] |
| | 0.3 | 100 miles away from a nobody jerk. you find a rip in space boarding a bus on west 96th [...] | 10 minutes in the past, before you were scheduled to meet a friend for coffee. You realize that by altering the past, [...] |
| | 0.7 | 10 years in the future. you are discovering all sorts of things. it comes to you [...] | 7 years ago in a Cafe you have been to before. [...] |
| | 1.0 | 10 years ago, trying to figure out where the next stride will take you or perhaps where [...] | 7 years ago in a Cafe you have been to before, supposed to meet with a friend that never showed up. [...] |
| Formality | 0.0 | 20! LOL that doesn't seem right or fair. [...] | in the past... [...] |
| | 0.3 | 7 years older. END OF STORY!! Things were going well until you started to take [...] | in the past, an absolute fantasy. You see a young version of yourself there, who [...] |
| | 0.7 | 40 or older, happily married, and with 3 kids. [...] | 10 minutes in the past, before the recent break-in at your apartment. [...] |
| | 1.0 | 21 years older from your prior adventures. [...] | 10 minutes in the past, before the recent break-in at your office. [...] |
| Simplicity | 0.0 | on a black public hillside, and a yellow sun butchered and bleeding in an ugly sky, and you know the cut of sandstone in the [...] | 20 years earlier, in a world before the great collapse. Children are playing, birds are chirping, and people are smiling. [...] |
| | 0.3 | 3 kilometres outside of town at a main road. you slowly move forward looking around your surroundings. Seeing a man sitting under a [...] | 7 years ago in a crucial moment of your past. [...] |
| | 0.7 | 300 years before your time. Some kids around you are running off to play in the forest. You stand there trying to figure out what to do [...] | 10 minutes in the past, before the recent downpour. How do you handle it? [...] |
| | 1.0 | 300 years back! It is 1828 in London. You stay in the alley until it becomes fully sunny. [...] | 10 minutes in the past, before the recent downpour of rains and flooding. [...] |
| Humor | 0.0 | 30 years in the past under another name. You're married to an old fling [...] | 7 years ago in a parking lot looking 7 years younger. [...] |
| | 0.3 | 30 years back, walking through the alleys. So much for not being surprised. [...] | 7 years ago in a parking lot looking 7 years younger. You see a car you can't remember [...] |
| | 0.7 | 75 years in the future, Washington DC's Newbridge Apartments has become an urban theme park [...] | 20 years earlier, in high school. Your younger self is looking at you, confused. You then see yourself in high school and [...] |
| | 1.0 | 255 years into the future, the day at the 'harmless' age of sixty million, a desperate, crazed - looking [...] | 10 minutes in the past, but you bring a hand-held portal weapon with you. [...] |

### A.9 Additional multi-dimensional plots

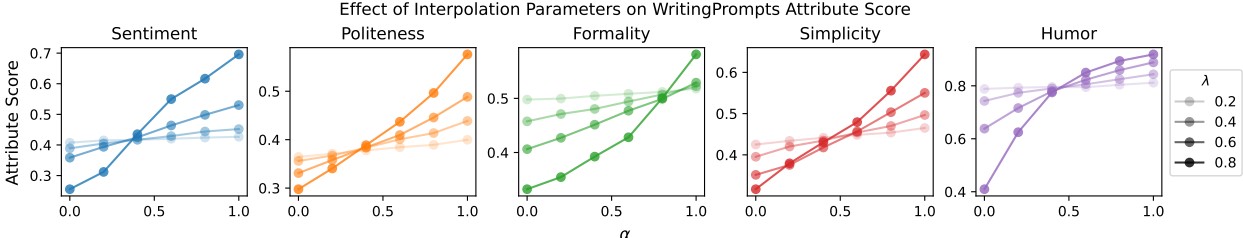

Figure 19: **Effect of $\alpha_i$ and $\lambda_i$ on 5-dimensional interpolation**. For each attribute, we show the attribute scores for models with the given $\alpha_i$ and $\lambda_i$ parameters, with all four other $\alpha_j = 1$ and $\lambda_j = (1 - \lambda_i)/4$. We find that increasing $\alpha_i$ consistently increases the attribute score and increasing $\lambda_i$ consistently increases the effect of $\alpha_i$.

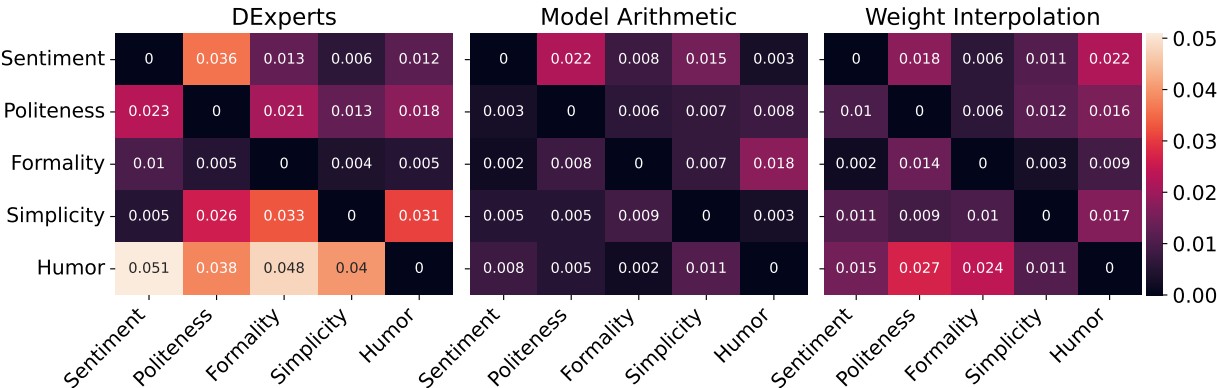

Figure 20: **Weight interpolation has entanglement lower than or comparable to prior approaches.** For each pair of dimensions, we fix (scaled) $\alpha_i = 0$ for the dimension in each row and vary (scaled) $\alpha_j$ between 0 and 1 for the dimension in each column. We set $\lambda_i = \lambda_j = 0.5$. Then, we report entanglement as the area under the curve of absolute value of change in attribute score as $\alpha_j$ increases. Weight interpolation is less entangled than DExperts (Liu et al., 2021) and has comparable entanglement to model arithmetic (Dekoninck et al., 2024).

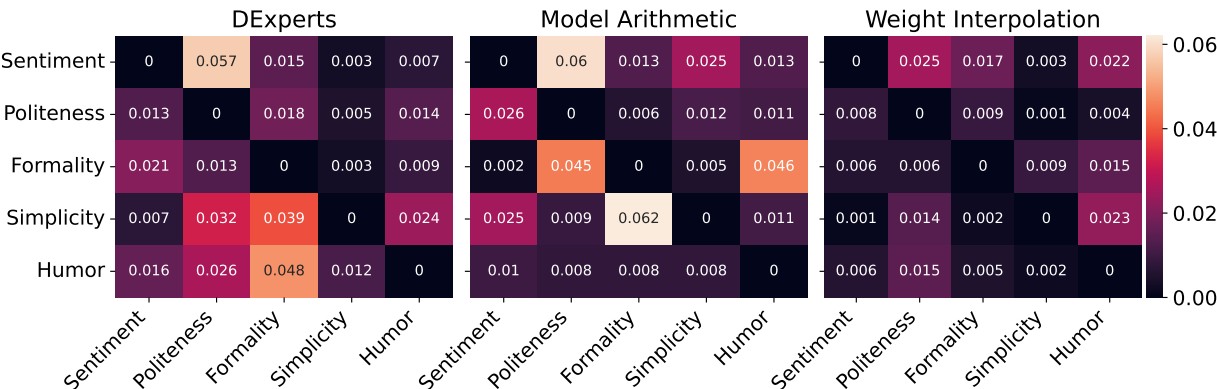

Figure 21: **Llama-2-13b weight interpolation has entanglement lower than or comparable to prior approaches.** For each pair of dimensions, we fix (scaled) $\alpha_i = 0$ for the dimension in each row and vary (scaled) $\alpha_j$ between 0 and 1 for the dimension in each column. We set $\lambda_i = \lambda_j = 0.5$. Then, we report entanglement as the area under the curve of absolute value of change in attribute score as $\alpha_j$ increases. Weight interpolation is less entangled than DExperts (Liu et al., 2021) and has comparable entanglement to model arithmetic (Dekoninck et al., 2024).

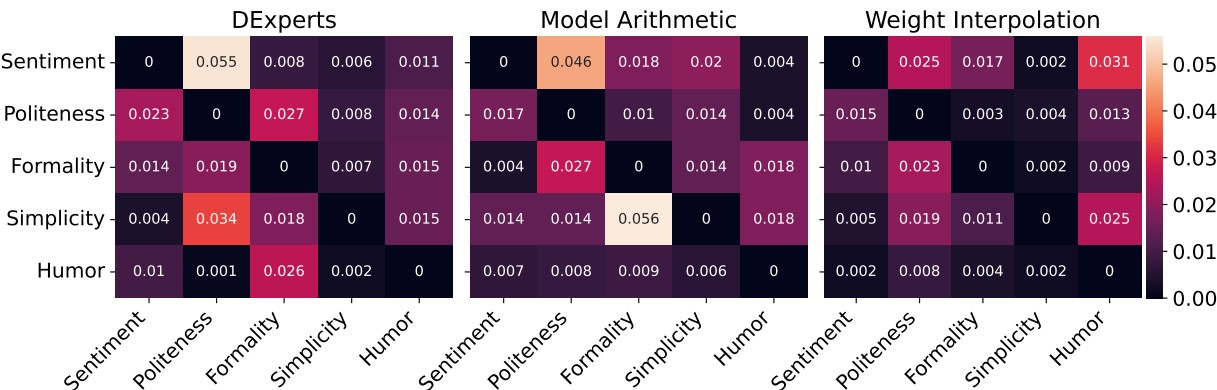

Figure 22: **Llama-2-13b weight interpolation has entanglement lower than or comparable to prior approaches.** For each pair of dimensions, we fix (scaled) $\alpha_i = 1$ for the dimension in each row and vary (scaled) $\alpha_j$ between 0 and 1 for the dimension in each column. We set $\lambda_i = \lambda_j = 0.5$. Then, we report entanglement as the area under the curve of absolute value of change in attribute score as $\alpha_j$ increases. Weight interpolation is less entangled than DExperts (Liu et al., 2021) and has comparable entanglement to model arithmetic (Dekoninck et al., 2024).

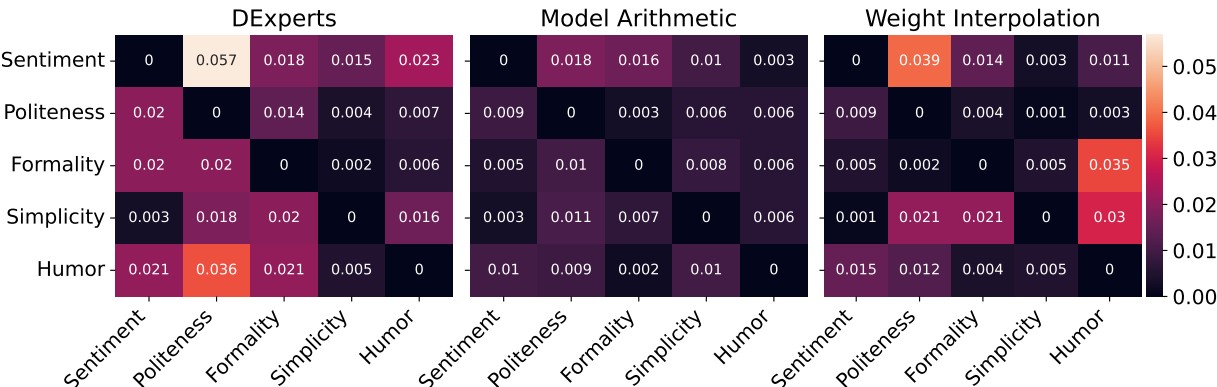

Figure 23: **Llama-3.1-8B weight interpolation has entanglement lower than or comparable to prior approaches.** For each pair of dimensions, we fix (scaled) $\alpha_i = 0$ for the dimension in each row and vary (scaled) $\alpha_j$ between 0 and 1 for the dimension in each column. We set $\lambda_i = \lambda_j = 0.5$. Then, we report entanglement as the area under the curve of absolute value of change in attribute score as $\alpha_j$ increases. Weight interpolation has comparable entanglement to DExperts (Liu et al., 2021) and model arithmetic (Dekoninck et al., 2024).

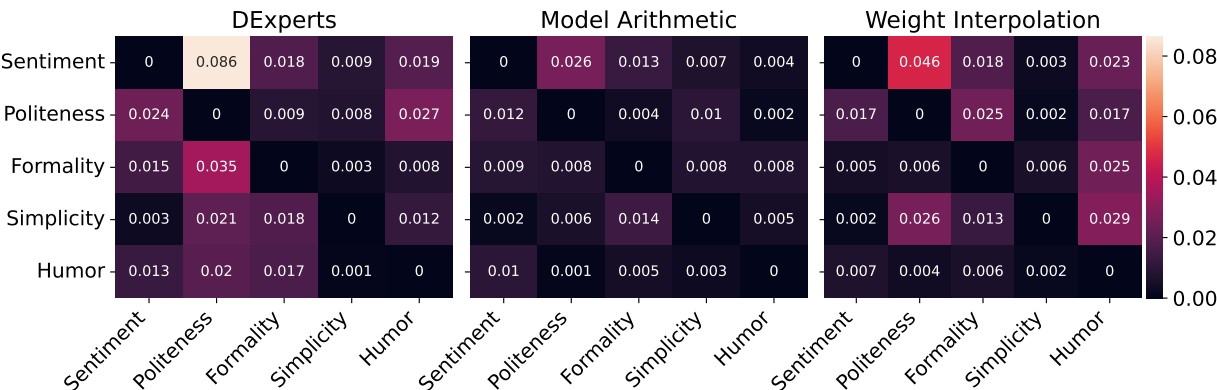

Figure 24: **Llama-3.1-8B weight interpolation has entanglement lower than or comparable to prior approaches.** For each pair of dimensions, we fix (scaled) $\alpha_i = 1$ for the dimension in each row and vary (scaled) $\alpha_j$ between 0 and 1 for the dimension in each column. We set $\lambda_i = \lambda_j = 0.5$. Then, we report entanglement as the area under the curve of absolute value of change in attribute score as $\alpha_j$ increases. Weight interpolation has comparable entanglement to DExperts (Liu et al., 2021) and model arithmetic (Dekoninck et al., 2024).

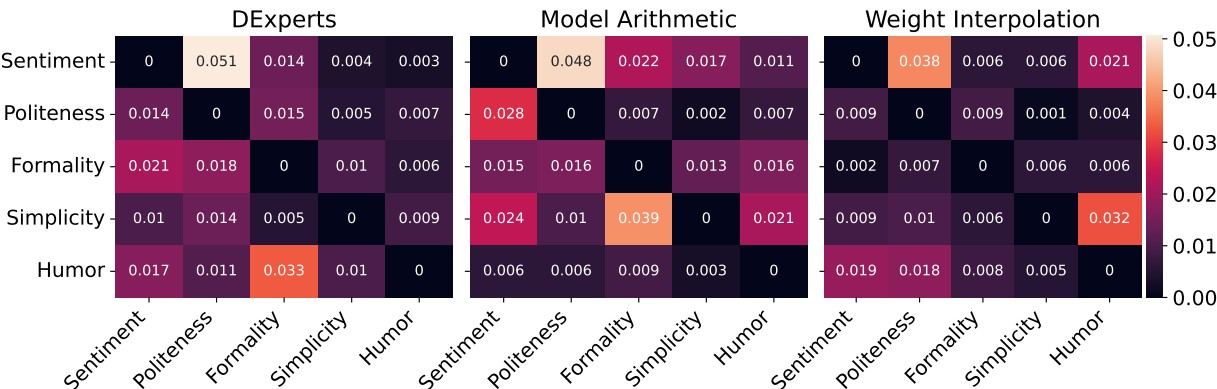

Figure 25: **Qwen3-8B-Base weight interpolation has entanglement lower than or comparable to prior approaches.** For each pair of dimensions, we fix (scaled) $\alpha_i = 0$ for the dimension in each row and vary (scaled) $\alpha_j$ between 0 and 1 for the dimension in each column. We set $\lambda_i = \lambda_j = 0.5$. Then, we report entanglement as the area under the curve of absolute value of change in attribute score as $\alpha_j$ increases. Weight interpolation has lower entanglement than DExperts (Liu et al., 2021) and model arithmetic (Dekoninck et al., 2024).

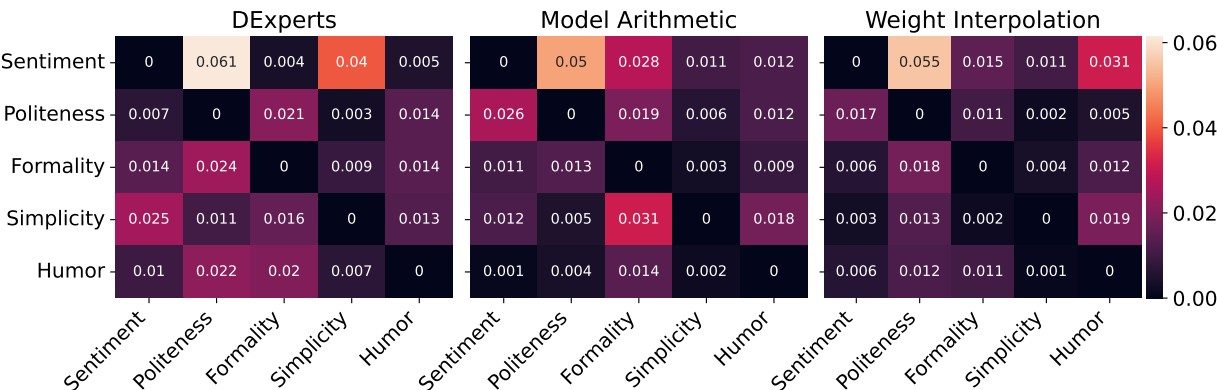

Figure 26: **Qwen3-8B-Base weight interpolation has entanglement lower than or comparable to prior approaches.** For each pair of dimensions, we fix (scaled) $\alpha_i = 1$ for the dimension in each row and vary (scaled) $\alpha_j$ between 0 and 1 for the dimension in each column. We set $\lambda_i = \lambda_j = 0.5$. Then, we report entanglement as the area under the curve of absolute value of change in attribute score as $\alpha_j$ increases. Weight interpolation is less entangled than DExperts (Liu et al., 2021) and has comparable entanglement to model arithmetic (Dekoninck et al., 2024).

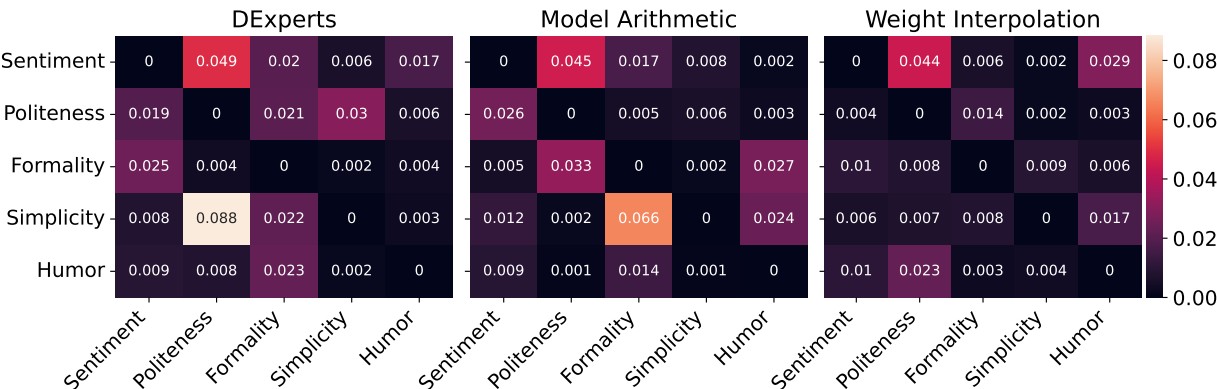

Figure 27: **Qwen3-14B-Base weight interpolation has entanglement lower than or comparable to prior approaches.** For each pair of dimensions, we fix (scaled) $\alpha_i = 0$ for the dimension in each row and vary (scaled) $\alpha_j$ between 0 and 1 for the dimension in each column. We set $\lambda_i = \lambda_j = 0.5$. Then, we report entanglement as the area under the curve of absolute value of change in attribute score as $\alpha_j$ increases. Weight interpolation has lower entanglement than DExperts (Liu et al., 2021) and model arithmetic (Dekoninck et al., 2024).

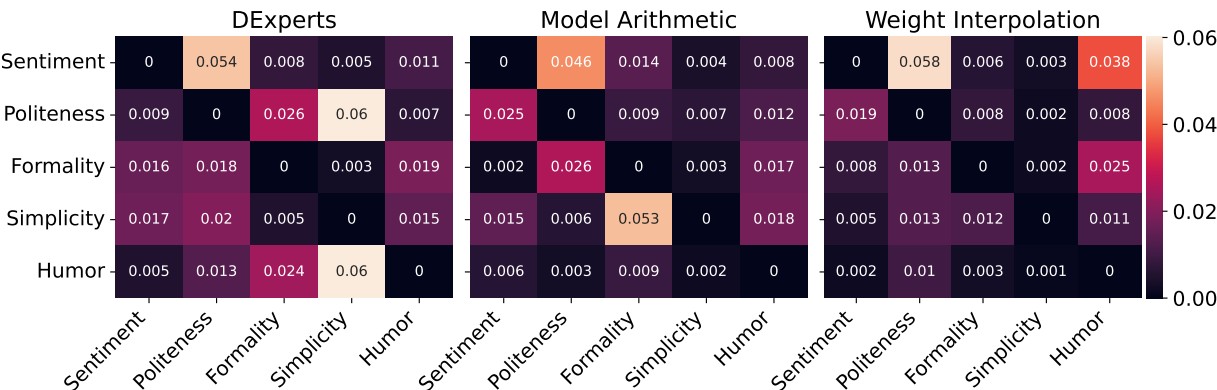

Figure 28: **Qwen3-14B-Base weight interpolation has entanglement lower than or comparable to prior approaches.** For each pair of dimensions, we fix (scaled) $\alpha_i = 1$ for the dimension in each row and vary (scaled) $\alpha_j$ between 0 and 1 for the dimension in each column. We set $\lambda_i = \lambda_j = 0.5$. Then, we report entanglement as the area under the curve of absolute value of change in attribute score as $\alpha_j$ increases. Weight interpolation is less entangled than DExperts (Liu et al., 2021) and has comparable entanglement to model arithmetic (Dekoninck et al., 2024).

### A.10 Additional multi-dimensional lambda simplex plots

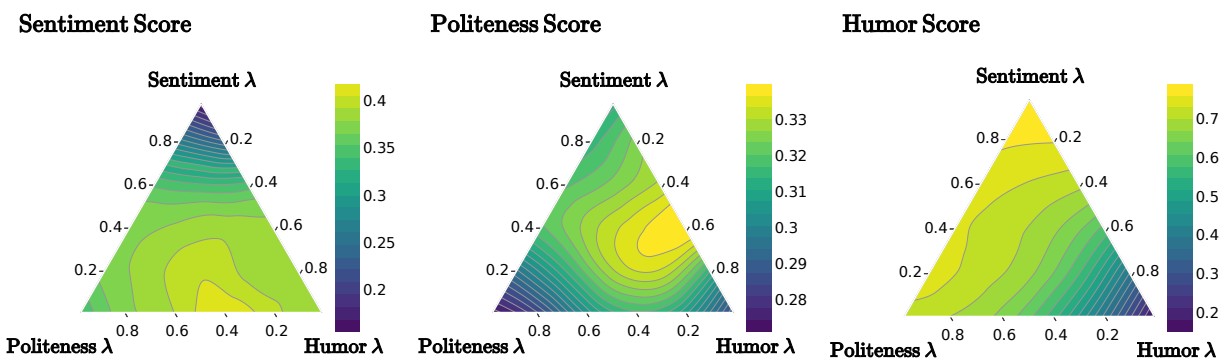

Figure 29: **Effect of $\lambda_i$ on interpolation between the sentiment, politeness, and humor dimensions for $\alpha_i = 0$.** The vertices of the triangle represent the models with $\alpha_i = 0$ for each of the three dimensions.

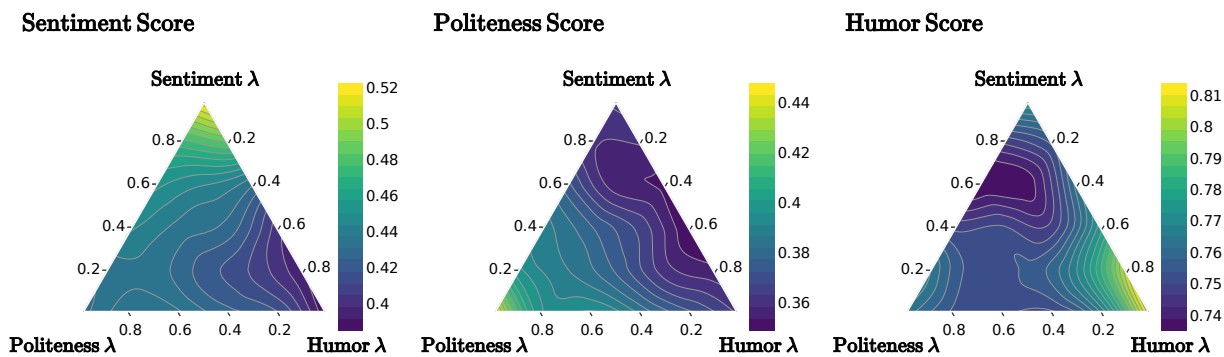

Figure 30: **Effect of $\lambda_i$ on interpolation between the sentiment, politeness, and humor dimensions for $\alpha_i = 0.5$.** The vertices of the triangle represent the models with $\alpha_i = 0.5$ for each of the three dimensions.

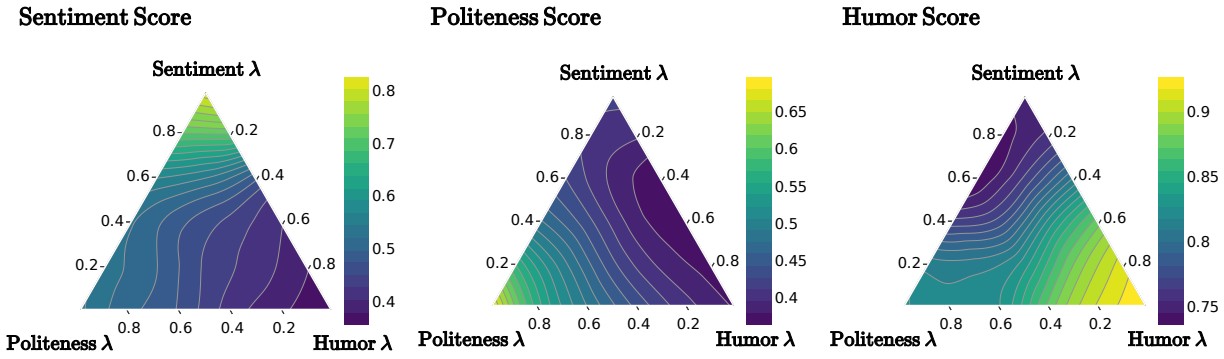

Figure 31: **Effect of $\lambda_i$ on interpolation between the sentiment, politeness, and humor dimensions for $\alpha_i = 1$.** The vertices of the triangle represent the models with $\alpha_i = 1$ for each of the three dimensions.

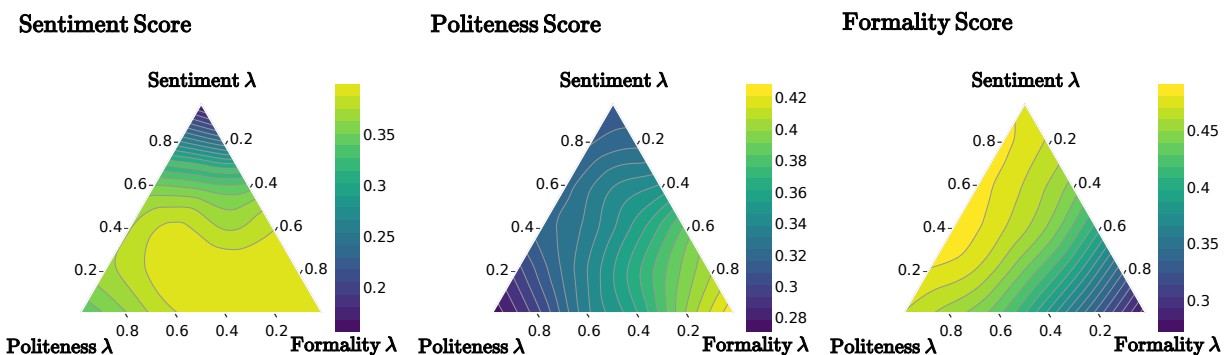

Figure 32: **Effect of $\lambda_i$ on interpolation between the sentiment, politeness, and formality dimensions for $\alpha_i = 0$.** The vertices of the triangle represent the models with $\alpha_i = 0$ for each of the three dimensions.

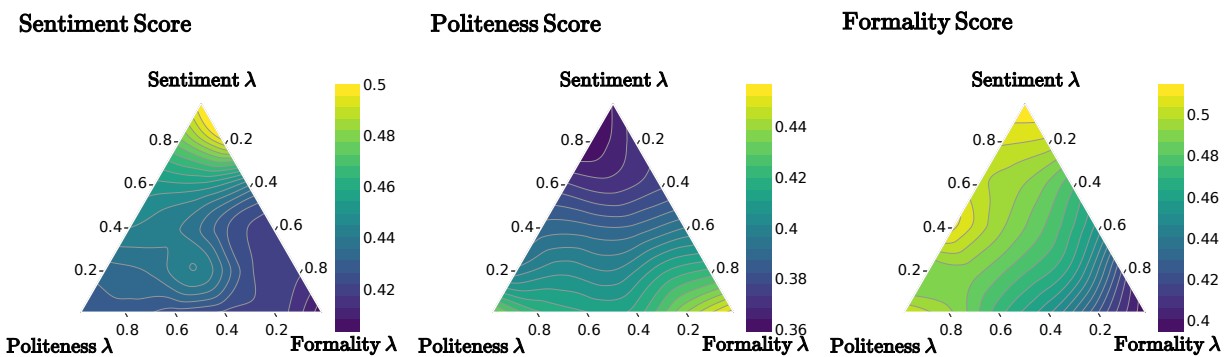

Figure 33: **Effect of $\lambda_i$ on interpolation between the sentiment, politeness, and formality dimensions for $\alpha_i = 0.5$.** The vertices of the triangle represent the models with $\alpha_i = 0.5$ for each of the three dimensions.

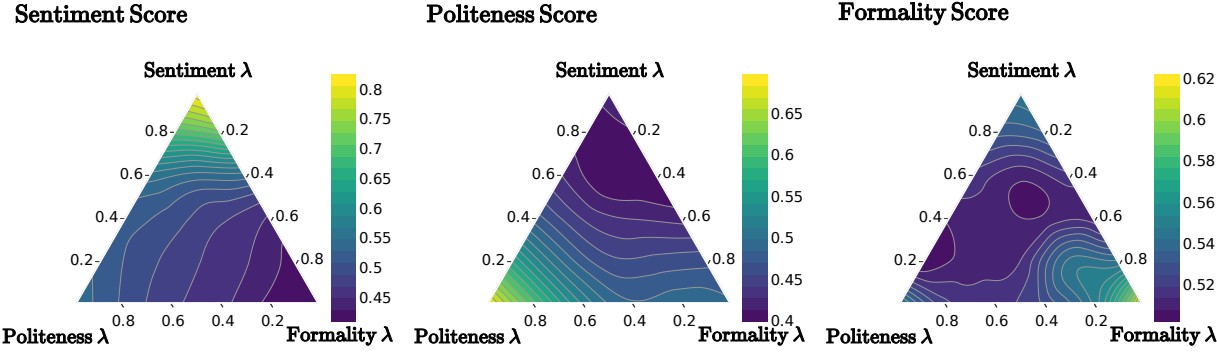

Figure 34: **Effect of $\lambda_i$ on interpolation between the sentiment, politeness, and formality dimensions for $\alpha_i = 1$.** The vertices of the triangle represent the models with $\alpha_i = 1$ for each of the three dimensions.

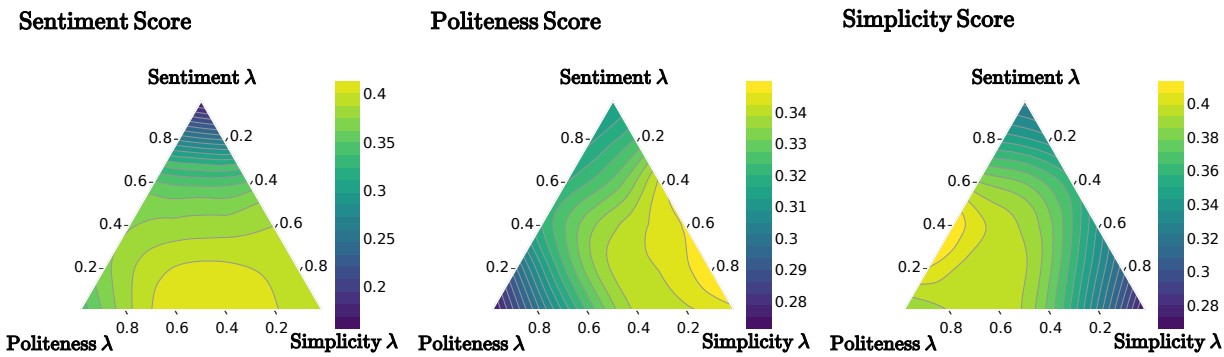

Figure 35: **Effect of $\lambda_i$ on interpolation between the sentiment, politeness, and simplicity dimensions for $\alpha_i = 0$.** The vertices of the triangle represent the models with $\alpha_i = 0$ for each of the three dimensions.

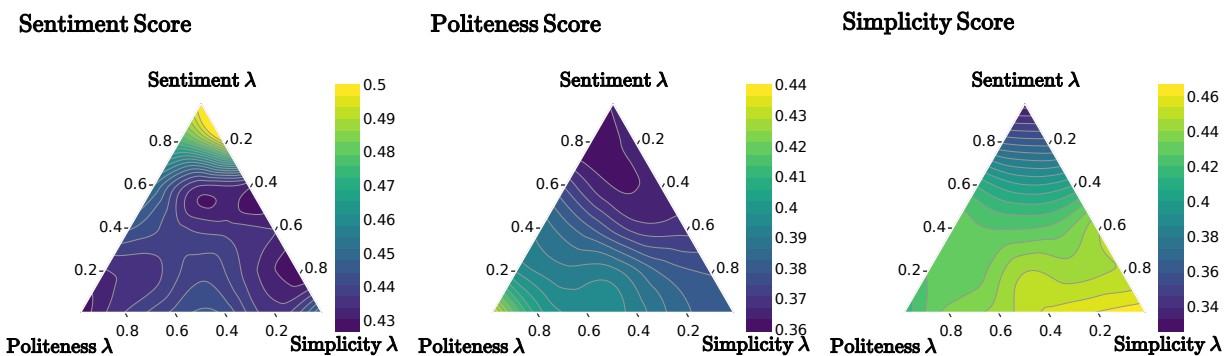

Figure 36: **Effect of $\lambda_i$ on interpolation between the sentiment, politeness, and simplicity dimensions for $\alpha_i = 0.5$.** The vertices of the triangle represent the models with $\alpha_i = 0.5$ for each of the three dimensions.

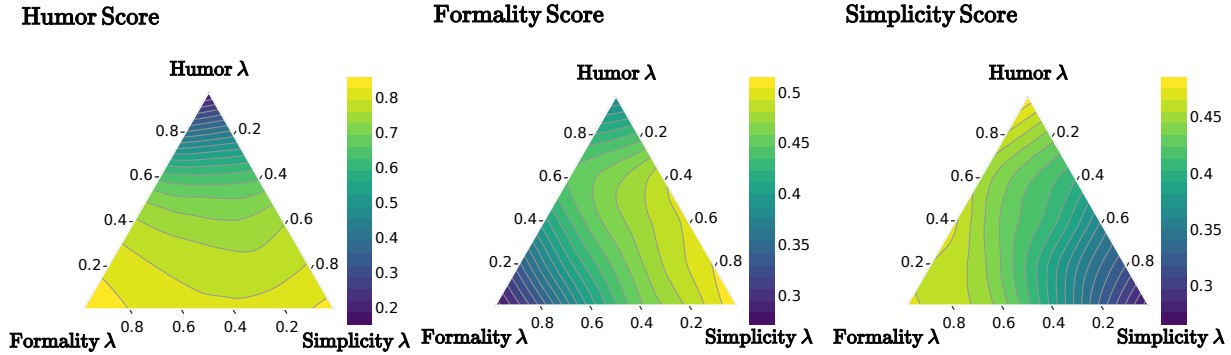

Figure 37: **Effect of $\lambda_i$ on interpolation between the humor, formality, and simplicity dimensions for $\alpha_i = 0$.** The vertices of the triangle represent the models with $\alpha_i = 0$ for each of the three dimensions.

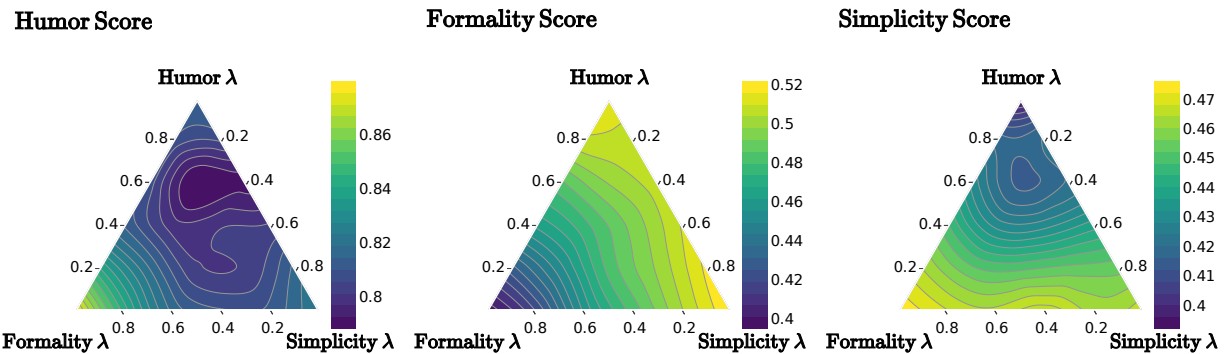

Figure 38: **Effect of $\lambda_i$ on interpolation between the humor, formality, and simplicity dimensions for $\alpha_i = 0.5$.** The vertices of the triangle represent the models with $\alpha_i = 0.5$ for each of the three dimensions.

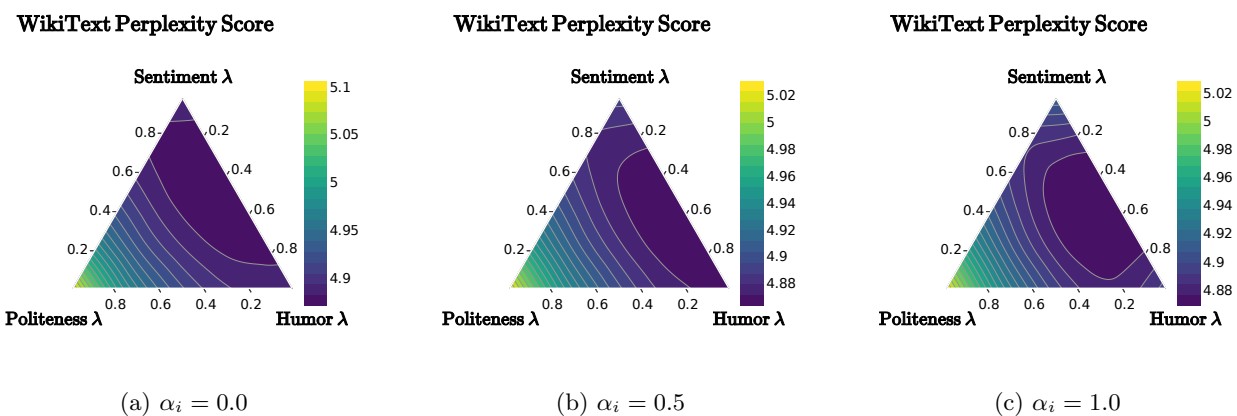

(a) $\alpha_i = 0.0$        (b) $\alpha_i = 0.5$        (c) $\alpha_i = 1.0$

Figure 39: **Effect of $\lambda_i$ on perplexity for interpolation between the sentiment, politeness, and humor dimensions for various $\alpha_i$ values.** The vertices of the triangle represent the models with the given $\alpha_i$ value for each of the three dimensions. The perplexity for each model is bounded above by the perplexities of the fine-tuned anchor models.

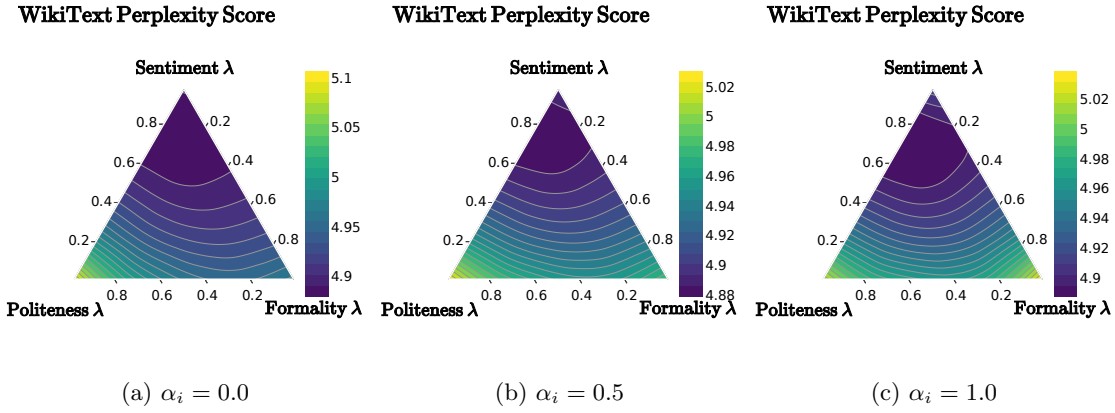

(a) $\alpha_i = 0.0$        (b) $\alpha_i = 0.5$        (c) $\alpha_i = 1.0$

Figure 40: **Effect of $\lambda_i$ on perplexity for interpolation between the sentiment, politeness, and formality dimensions for various $\alpha_i$ values.** The vertices of the triangle represent the models with the given $\alpha_i$ value for each of the three dimensions. The perplexity for each model is bounded above by the perplexities of the fine-tuned anchor models.

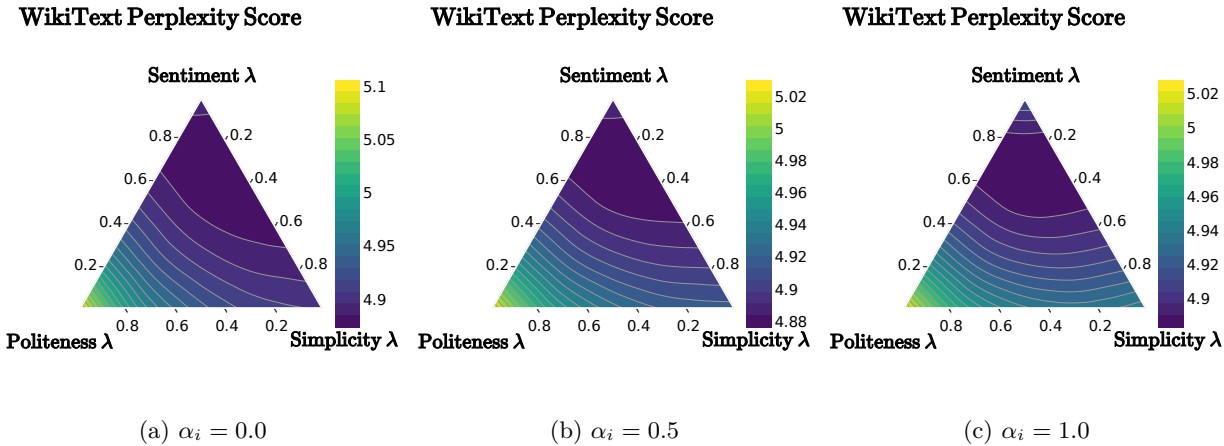

(a) $\alpha_i = 0.0$       (b) $\alpha_i = 0.5$       (c) $\alpha_i = 1.0$

Figure 41: **Effect of $\lambda_i$ on perplexity for interpolation between the sentiment, politeness, and simplicity dimensions for various $\alpha_i$ values.** The vertices of the triangle represent the models with the given $\alpha_i$ value for each of the three dimensions. The perplexity for each model is bounded above by the perplexities of the fine-tuned anchor models.

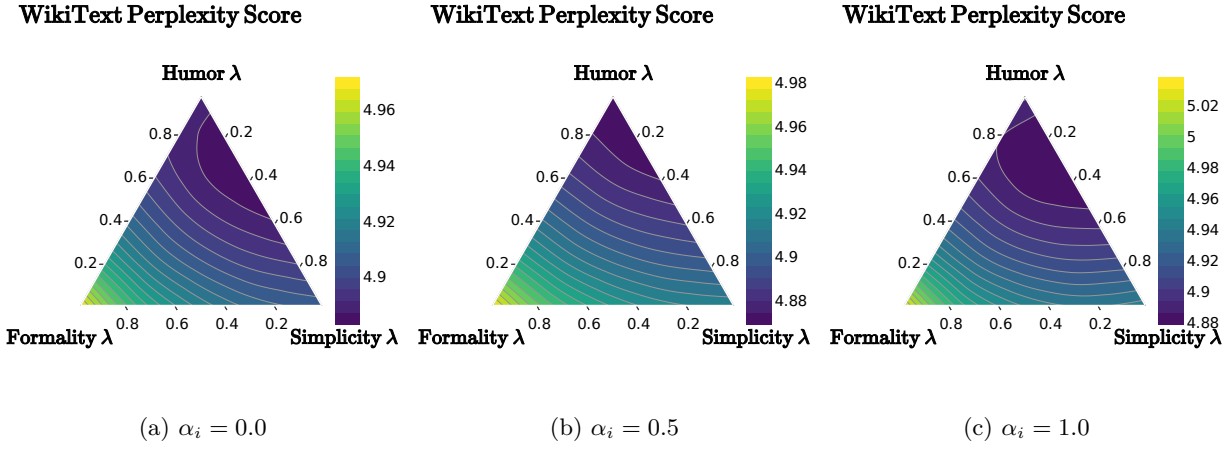

(a) $\alpha_i = 0.0$       (b) $\alpha_i = 0.5$       (c) $\alpha_i = 1.0$

Figure 42: **Effect of $\lambda_i$ on perplexity for interpolation between the humor, formality, and simplicity dimensions for various $\alpha_i$ values.** The vertices of the triangle represent the models with the given $\alpha_i$ value for each of the three dimensions. The perplexity for each model is bounded above by the perplexities of the fine-tuned anchor models.

### A.11 Multi-dimensional scaling (MDS) analysis of fine-tuned models

We project the weights of the LoRA fine-tuned endpoint models, as well as some of the interpolated models, into two dimensions using multi-dimensional scaling (MDS). As shown in Figure 43, we find that the interpolating between the endpoint fine-tuned models generally results in models that are closer to the base model. This is expected behavior since we would anticipate that the base model is fairly neutral with respect to all attributes.

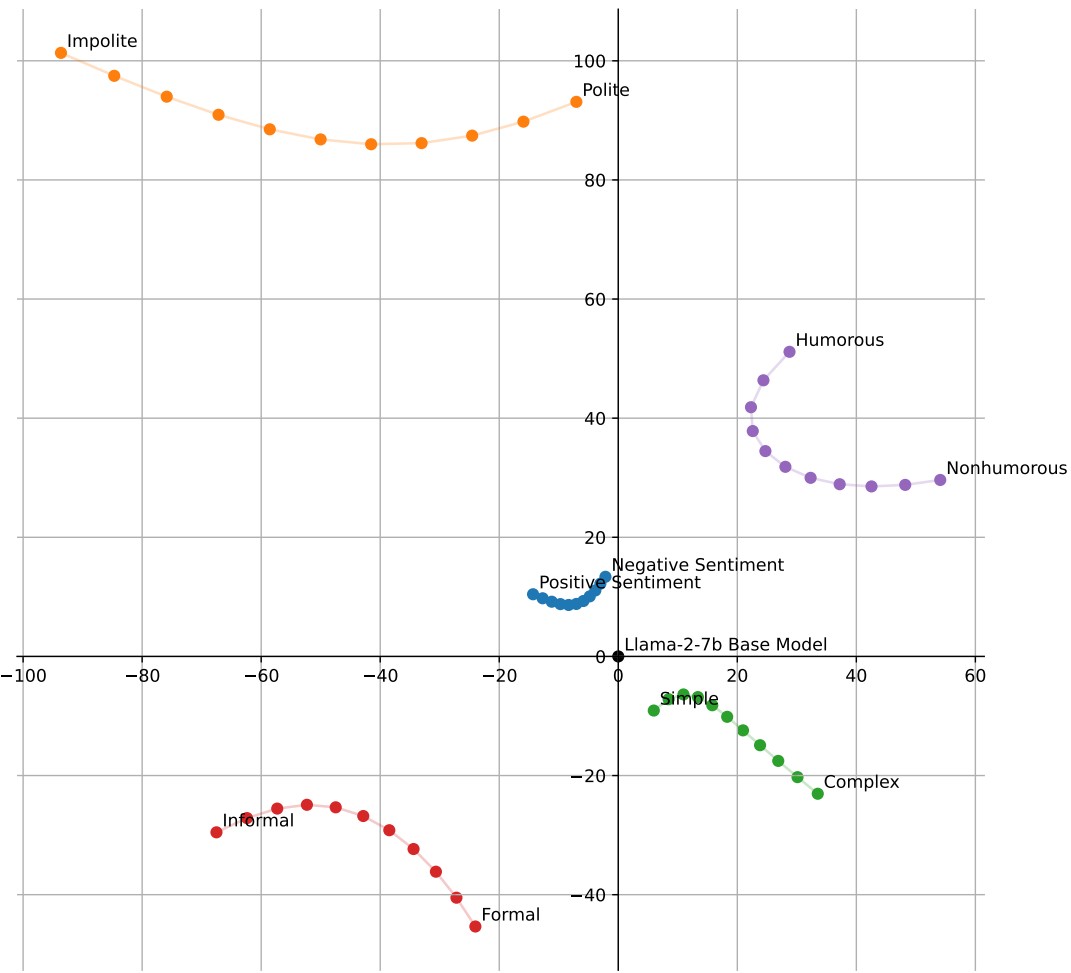

Figure 43: **Multi-dimensional scaling (MDS) plot for the fine-tuned models and linear interpolations.** This plot shows the 2-dimensional MDS projection of the fine-tuned anchor models and the models interpolated at intervals of 0.1. This corresponds to the models in Figure 2.

### A.12 Similarity between pairs of fine-tuned models

We compute the cosine similarities between LoRA weights in Figure 44 and use the average squared L2 norm of the difference between the LoRA updates to analyze the distances between models in Figure 45. For the cosine similarity, the models appear relatively orthogonal, except for the opposing models from the same attribute dimension. In terms of L2 norm, the models fine-tuned on the classes with attribute score of 1 (positive sentiment, polite, simple, formal, humorous) tend to be closer to the other models than the models fine-tuned on classes with attribute score 0. We also find that the polite and impolite LoRA fine-tuned endpoint models are the farthest from the other models on average. This is consistent with the results from the MDS plot (Figure 43).

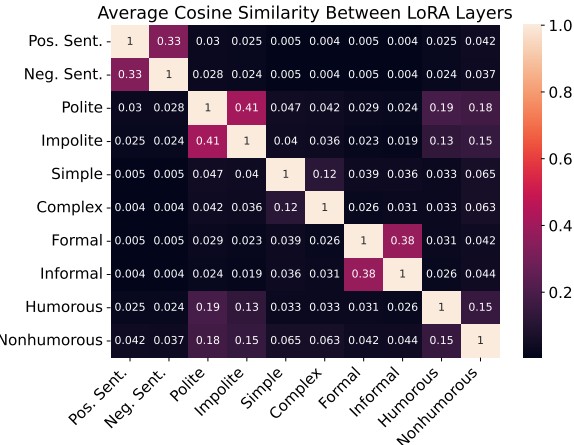

Figure 44: **Cosine similarity of LoRA weights averaged across layers between each pair of fine-tuned anchor models.** The LoRA weights are all relatively orthogonal to each other, except some of the two endpoint models for the same attribute are less orthogonal to each other, as well as the politeness and humorous models.

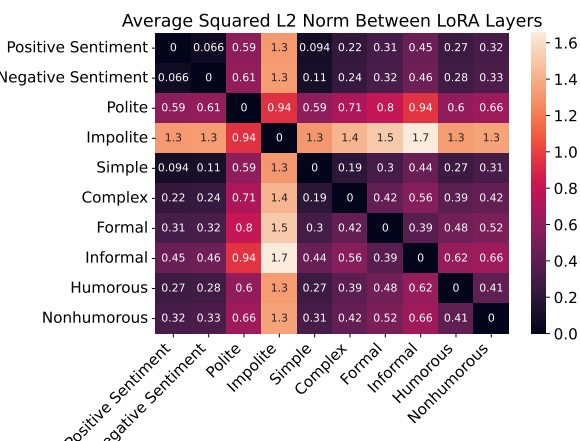

Figure 45: **Average pairwise squared L2 norms between LoRA layers.** The fine-tuned anchor models trained on the class with attribute score of 1 tend to be closer to the other models than those trained on the class with attribute score of 0. The polite and impolite models are the farthest from the other models.

