# OpenReview forum: "Continuous Language Model Interpolation yields Dynamic and Controllable Text Generation"
_TMLR — Accepted by TMLR_

### Review · Reviewer_JTFg · 2025-06-06

**Summary Of Contributions:**

The paper introduces a method for controlling certain attributes of LLM outputs (e.g., politeness, humor) by averaging the weights of model variants fine-tuned to increase or decrease each individual attribute. The method relies on two types of parameters: first, parameters that independently weigh each attribute's level, and second, parameters that weigh attributes relative to one another (i.e., forming a simplex). The authors conduct a series of experiments to evaluate their method against two baselines. They first focus on changes to individual attributes independently, demonstrating that linear interpolation successfully increases or decreases each target attribute. The authors also show that extrapolation beyond the weights of the two fine-tuned endpoint models is possible, though it quickly leads to instability and less predictable outputs. Finally, they demonstrate that adjusting the relative weights of attributes does increase the presence of the respective attribute in the model's responses, with some (but limited) interference between different attributes and their corresponding parameters.

**Audience:**

Yes

**Broader Impact Concerns:**

The work already contains a broader impact statement that sufficiently discusses its ethical implications.

**Claims And Evidence:**

Yes

**Requested Changes:**

Here are some more specific comments regarding the paper:

* In terms of writing, I found the introduction of the paper somewhat hard to follow. To me, it only made sense after I had started reading section 2. The authors start talking about "distinct and continuously evolving preferences of users" without explaining via concrete examples what they mean by "preferences". Then they briefly argue that user preferences are continuous before jumping into a detailed discussion of prior work on controllable text generation, which makes the argument regarding continuous preferences (i.e., the main point of the paper) quite weak. Specifically, the phrase "expressing fine-grained continuous preferences (e.g., simplifying a response by 25%) is often difficult in---inherently discrete---natural language" is particularly confusing. It is unclear how the fact that natural language is discrete is relevant to the argument.

* The experimental evaluation depends on a single model (Llama2-7b). It would be interesting to see whether the effectiveness of linear weight interpolation changes depending on (i) the model size and (ii) different families of models. This is missing from the current version of the paper.

* In equation 2, I believe that the term $\theta_\text{PRE}$ is unnecessary in the first equation since its appearance in the second equation would result by the convex combination of $\theta_{+i}$ and $\theta_{-i}$.

* In section 2.4, the authors mention that the two baselines require 2*number of dimensions + 1 inference passes, but it is not immediately clear why. More generally, since the authors compare with only 2 baselines, I believe the paper would benefit from a better description of those two baselines. What exactly do they do? I believe that the baseline "model arithmetic" is just a different way of prompting the model to generate responses with the required attributes, so it would be quite easy to describe the method in words to provide more context.

* In figure 2, the baseline "DExperts" seems to behave similarly to the method that the authors propose. Again, some clarity regarding what this baseline does would help, but I think it would also be useful to emphasize more what are the benefits of the proposed method in comparison to this baseline.

* It is a bit secondary to the main topic of the paper, but it would be useful if the authors could look into what attribute of the network itself leads to extrapolation not working, rather than just reporting properties of the output (i.e., perplexity). One idea could be to look into the norms of the network's weights (e.g., the spectral norm) and how these vary as a function of $\alpha$.

* In section 3.1, the authors write that interpolating the weights of fine-tuned versions of a model leads to controllable output when considering *up to five* attribute dimensions. Is there some problem with the method when considering more than five attributes? If so, could the authors add relevant experimental results?

* In Figure 7 (left), the humor score seems to increase comparably when increasing the parameter $\lambda$ for both humor and formality. Does that imply that increasing the parameter for formality also causes an increase in humor? That's a bit strange since one would expect outputs with humor to be less formal. Related to that, in Figure 8, shouldn't there be an entanglement between formality and humor? Can the authors clarify this?

* The discussion of related work in section 4.1 is a bit shallow, with relatively uninformative descriptions of what other methods are doing. Can the authors explain what makes the adaptation of those methods to the multiple attribute setting difficult or not possible?

* In terms of limitations, something that the authors do not mention is that in order for their method to work, one needs to perform fine-tuning twice for each of the attributes under consideration, which presents a significant overhead compared with approaches that control the outputs via prompting, for example. I think this needs to be highlighted.

* In terms of future work, do the authors think that their method could be used to generalize to other attributes without requiring additional fine-tuning? For example, if you have attributes "humor" and "irony", I can imagine the interpolation of the fine-tuned models could lead to outputs that have a high score for an attribute "sarcasm".

**Strengths And Weaknesses:**

Strengths:
* The method is simple and easy to understand and implement, since the main idea is just linear interpolation of the weights of fine-tuned model variants
* The experimental evaluation and results are intuitive and effectively show that the method works as expected

Weaknesses:
* The authors could elaborate more on the advantages/disadvantages of the proposed method in comparison with related work
* The experimental evaluation seems to rely on a single LLM

---

> ### Author Response · Authors · 2025-08-01
> **Response to reviewer JTFg**
>
> We thank the reviewer for their comments and appreciate their positive feedback about the effectiveness of the experiments and the easy implementation of the method.
>
> We have revised the introduction for clarity and corrected the typo in Equation 2 found by the reviewer.
>
> > The experimental evaluation depends on a single model.
>
> We have run the single attribute linear interpolation results in Figure 2 and Table 1 and the entanglement analysis in Figure 8 for 4 additional models (Sections A.7 and A.7).
>
> > The two baselines require 2*number of dimensions + 1 inference passes, but it is not immediately clear why.
>
> We have added a more detailed explanation of the DExperts and model arithmetic baselines (section 2.4). In short, DExperts uses a base model and 2 fine-tuned endpoint models for each attribute. Given a prompt $x_{<t}$, if we denote the output logits of the base model at time $t$ as $z_t$ and the logits of the two endpoint models as $z_t^+$ and $z_t^-$, respectively, then the output probability distribution is defined by
> $$P(X_t | x_{<t}) = \text{softmax}(z_t + \alpha(z_t^+ - z_t^-))$$
> To compute the DExperts generations, we need to compute the logits for the base model and 2 * number of dimensions endpoint models, resulting in 2*number of dimensions + 1 inference passes. The model arithmetic baseline uses the same formula for computing the output probability distribution, but instead of using 2 fine-tuned endpoint models, it uses 2 prompt-conditioned endpoint models.
>
> > In figure 2, the baseline "DExperts" seems to behave similarly to the proposed method
>
> We have emphasized the benefits of our approach compared to DExperts in Section 2.4 of the revised paper. The main benefit of our method in comparison to DExperts is that our approach only requires a single inference pass, making it much more scalable.
>
> > What attribute of the network itself leads to extrapolation not working
>
> We thank the reviewer for this suggestion and have added additional MDS analysis to Appendix Section A.11.
>
> > Is there some problem with the method when considering more than five attributes?
>
> We selected five attributes because they correspond to the primary style dimensions with publicly available, high-quality labeled datasets suitable for fine-tuning [1]. There is nothing in principle that limits our method to five attributes, but evaluating more than five attributes would require access to additional high-quality datasets, which is currently constrained by practical considerations. We view this as a promising direction for future work and expect our approach to generalize well to a broader set of attributes.
>
> > In Figure 7 (left), the humor score seems to increase
>
> In this case, there is a difference between entanglement in terms of $\lambda$ and $\alpha$. As mentioned in Section 3.2.1, there is an increase in humor score when the $\lambda$ for formality increases in Figure 7 because the formal endpoint model has a high humor score, so the mixture of models is the most neutral model. As shown in Figure 25 (left), the informal endpoint model also has a high humor score, which indicates that the entire GYAFC dataset [2] used to fine-tune the formal and informal models is correlated with humorous text.
>
> However, the analysis of $\alpha$ entanglement in Figure 8 shows that as $\alpha$ changes for formality, the humor score does not change. As a result, in contrast to taking different weighted averages of the humorous and formal models, moving between the formal and informal endpoint models does not significantly effect the humor score.
>
> > The discussion of related work in section 4.1 is a bit shallow
>
> The central challenge of adapting these approaches to our setting is not extending to multiple attributes, but rather to the fine-grained setting where we are interested in controlling outputs in the entire space of potential attribute strength combinations. The majority of these approaches (other than DExperts and model arithmetic) optimize for a fixed set of controls criteria, which means that they would need to be re-run for each combination of attribute interpolation parameters. We clarify this point in the related work section.
>
> > In terms of limitations, something that the authors do not mention is that in order for their method to work, one needs to perform fine-tuning twice for each of the attributes under consideration.
>
> We thank the reviewer for this feedback and have mentioned this limitation in Section 2.4 and the related work section of the revised paper.
>
> > In terms of future work, do the authors think that their method could be used to generalize to other attributes without requiring additional fine-tuning?
>
> We agree that this would be an interesting direction for future work but is out of the scope of this paper.
>
> [1] Di Jin et. al. Deep Learning for Text Style Transfer: A Survey. Computational Linguistics 2022.
> [2] Sudha Rao et. al. Dear sir or madam, may I introduce the GYAFC datasetr. NAACL 2018.

---

### Review · Reviewer_YfNs · 2025-06-09

**Summary Of Contributions:**

This paper presents a framework for continuous and controllable text generation via linear interpolation of LoRA-adapted LLMs. For each stylistic attribute (e.g., simplicity, formality), the authors fine-tune two anchor models on opposite extremes, then interpolate between them using per-attribute weights. By mixing across attributes, the method parameterizes a convex hull of controllable models. Experiments on Llama2-7B across five style attributes show smooth, predictable control with minimal attribute entanglement, outperforming DExperts and model arithmetic in precision and efficiency.

**Audience:**

Yes

**Broader Impact Concerns:**

None.

**Claims And Evidence:**

Yes

**Requested Changes:**

- Extending the analysis to more model families.

- Even a small-scale human study or readability/diversity metrics would complement the classifier-based scores and confirm perceptual control.

**Strengths And Weaknesses:**

## Strengths:

- The idea of using LoRA-based weight interpolation to span a continuous control space is elegant and practical.

- The method supports simultaneous control of several style attributes, outperforming prior methods in precision and controllability.

- Experiments across five attributes show strong trends, low entanglement, and controlled behavior, supported by clear visualizations (e.g., simplex plots, entanglement metrics).

- The paper is well-organized, with helpful figures (especially Figure 1) and detailed methodology.

## Weaknesses:

- Only one base model (LLaMA2-7B) with five style attributes is tested. It's unclear whether the approach generalizes to other models, e.g., newer model families like Qwen-2.5 and Llama-3.

- Attribute strength is measured using RoBERTa-based classifiers. While standard, this risks overestimating control quality without human validation.

- Outside the [0,1] interpolation range, generation quality degrades, limiting usable control space.

---

> ### Author Response · Authors · 2025-08-01
> **Response to reviewer YfNs**
>
> We thank the reviewer for their feedback and positive comments about how the method is elegant and practical and how experiments show control over multiple attributes simultaneously.
>
> **Response to weaknesses:**
>
> > Only one base model (LLaMA2-7B) with five style attributes is tested. It's unclear whether the approach generalizes to other models, e.g., newer model families like Qwen-2.5 and Llama-3.
>
> We have run the single attribute linear interpolation results in Figure 2 and Table 1 and the entanglement analysis in Figure 8 for 4 additional models: Llama-2-13b-hf, Llama-3.1-8B, Qwen3-8B-Base, and Qwen3-14B-Base. We add the new results to Sections A.7 and A.9 in the appendix of the revised paper.
>
> We find that across different model sizes and model families, linear weight interpolation most closely follows the attribute scores of the ground truth fine-tuned models. In the extended entanglement analysis, we find that for all of the models, linear weight interpolation has lower entanglement than DExperts and similar entanglement to model arithmetic. Furthermore, larger models have slightly lower entanglement on average.
>
> > Attribute strength is measured using RoBERTa-based classifiers. While standard, this risks overestimating control quality without human validation.
>
> We have added additional text quality analysis by computing the compression ratio, homogenization score, and self repetition score for the single attribute linear interpolation for all models in Appendix Table 6.  We find that for single attribute control, weight interpolation produces similar or higher quality text than previous approaches.
>
> > Outside the [0,1] interpolation range, generation quality degrades, limiting usable control space.
>
> We agree that the usable control space of the outputs is limited by the fine-tuned endpoints. However, comparable methods such as DExperts and model arithmetic also begin producing poor quality output in some of the extrapolation region. We leave a full comparison of stable extrapolation regions for different methods to future work.
>
> **Response to requested changes**
>
> > Extending the analysis to more model families.
>
> Please see the response to the first weakness above.
>
> > Even a small-scale human study or readability/diversity metrics would complement the classifier-based scores and confirm perceptual control.
>
> Please see response to the second weakness above.

---

### Review · Reviewer_zLdD · 2025-07-03

**Summary Of Contributions:**

The paper presents a method called continuous linear weight interpolation to blend LoRA fine-tuned models. It starts by training two "anchor" models for each style attribute—like simplicity, formality, or sentiment—representing opposite extremes.

A two-parameter system controls the blending: αᵢ adjusts how much to interpolate between anchor models for each attribute, while λᵢ controls each attribute's overall influence on the final model.

This results in a smooth, continuous space of models spanning the convex hull of all anchor models, enabling fine-grained control over multiple stylistic features.

This paper addresses key gaps in controllable text generation by enabling continuous, real-time control over multiple text attributes through simple weight interpolation. Its core innovation is modeling the space of all attribute combinations as a navigable convex hull, allowing users to fine-tune outputs (e.g., "70% formal, 30% humorous") without retraining. Unlike prior work that relied on discrete categories or complex ensembles, this approach reframes CTG as a smooth mathematical navigation problem.

**Audience:**

Yes

**Claims And Evidence:**

Yes

**Requested Changes:**

Acknowledge the limitation in the experimentation results, like only Llama2-7B is tested.

**Strengths And Weaknesses:**

# Strength

-  Identifies a clear λ threshold for when attribute interference becomes problematic.
-  Uses perplexity, attribute accuracy, and correlation analysis for a well-rounded view.
-  Exhaustively evaluates all 26 subsets of 5 attributes.

# Weaknesses
- Results are only shown for Llama2-7B, not other models.
- Relies entirely on RoBERTa classifiers for attribute control.

---

> ### Author Response · Authors · 2025-08-01
> **Response to reviewer zLdD**
>
> We thank the reviewer for their comments and positive feedback about the paper's well-rounded view and exhaustive evaluation.
>
> **Response to weaknesses**
> > Results are only shown for Llama2-7B, not other models.
>
> We have run the single attribute linear interpolation results in Figure 2 and Table 1 and the entanglement analysis in Figure 8 for 4 additional models: Llama-2-13b-hf, Llama-3.1-8B, Qwen3-8B-Base, and Qwen3-14B-Base. We add the new results to Sections A.7 and A.9 in the appendix of the revised paper.
>
> We find that across different model sizes and model families, linear weight interpolation most closely follows the attribute scores of the ground truth fine-tuned models. In the extended entanglement analysis, we find that for all of the models, linear weight interpolation has lower entanglement than DExperts and similar entanglement to model arithmetic. Furthermore, larger models have slightly lower entanglement on average.
>
> > Relies entirely on RoBERTa classifiers for attribute control.
>
> We have added additional text quality analysis by computing the compression ratio, homogenization score, and self repetition score for the single attribute linear interpolation for all models in Appendix Table 6.  We find that for single attribute control, weight interpolation produces similar or higher quality text than previous approaches.
>
> **Response to requested changes**
> > Acknowledge the limitation in the experimentation results, like only Llama2-7B is tested.
>
> Please see response to the first weakness above.

---

### Comment · Action_Editor_fKFs · 2025-07-08
**Discussion period**

Dear authors and reviewers,

Thank you all for taking part in the review process so far. As the discussion period for this submission has began, I encourage the authors to carefully read the reviews and submit a detailed response to each of them. The authors are also encouraged to directly revise the draft as necessary. Both authors and reviewers are encouraged to follow up with further discussions as they see fit.

Thank you very much and looking forward to productive discussion

---

### Decision · Action_Editor_fKFs · 2025-08-22

**Recommendation:** Accept as is

**Audience:**

Yes

**Audience Explanation:**

This paper will be of interest to researchers and practitioners working on controllable text generation, LoRA fine-tuning, and model merging. The work provides a detailed empirical characterization of a simple, practical, and efficient technique for controlling multiple generation attributes simultaneously.

**Claims And Evidence:**

Yes

**Claims Explanation:**

The paper investigates continuous linear interpolation of LoRA models as a method for dynamic and controllable text generation. The initial concerns brought up by reviewers were about using only a single base model and fully relying on roberta-based classifier for attribute control evaluation. The authors have addressed these in their revision by experimenting with multiple different models in the Llama and Qwen families, and adding more evaluation metrics.